# Single-zinc vacancy unlocks high-rate $H_2O_2$ electrosynthesis from mixed dioxygen beyond Le Chatelier principle

Qi Huang[1], Baokai Xia[1], Ming Li[1], Hongxin Guan[1], Markus Antonietti [2] & Sheng Chen [1,2] ✉

Le Chatelier's principle is a basic rule in textbook defining the correlations of reaction activities and specific system parameters (like concentrations), serving as the guideline for regulating chemical/catalytic systems. Here we report a model system breaking this constraint in $O_2$ electroreduction in mixed dioxygen. We unravel the central role of creating single-zinc vacancies in a crystal structure that leads to enzyme-like binding of the catalyst with enhanced selectivity to $O_2$, shifting the reaction pathway from Langmuir-Hinshelwood to an upgraded triple-phase Eley-Rideal mechanism. The model system shows minute activity alteration of $H_2O_2$ yields (25.89~24.99 mol $g_{cat}^{-1}$ $h^{-1}$) and Faradaic efficiencies (92.5%~89.3%) in the $O_2$ levels of 100%~21% at the current density of 50~300 mA $cm^{-2}$, which apparently violate macroscopic Le Chatelier's reaction kinetics. A standalone prototype device is built for high-rate $H_2O_2$ production from atmospheric air, achieving the highest Faradaic efficiencies of 87.8% at 320 mA $cm^{-2}$, overtaking the state-of-the-art catalysts and approaching the theoretical limit for direct air electrolysis (~345.8 mA $cm^{-2}$). Further techno-economics analyses display the use of atmospheric air feedstock affording 21.7% better economics as comparison to high-purity $O_2$, achieving the lowest $H_2O_2$ capital cost of 0.3 \$ $Kg^{-1}$. Given the recent surge of demonstrations on tailoring chemical/catalytic systems based on the Le Chatelier's principle, the present finding would have general implications, allowing for leveraging systems "beyond" this classical rule.

Two-electron oxygen reduction reaction (ORR) offers an alternative/supplementary route for traditional anthraquinone process, enabling the distributed on-demand production of $H_2O_2$ under ambient condition[1–3]:

$$O_2 + H_2O + 2e^- \rightarrow HO_2^- + OH^- \quad (E^o = 0.06 \text{ V vs. SHE, in alkaline media}) \quad (1)$$

$$O_2 + 2H^+ + 2e^- \rightarrow H_2O_2 \quad (E^o = 0.68 \text{ V vs. SHE, in neutral/acidic media}) \quad (2)$$

Research into viable ORR catalytic systems has received growing interest[4–9], but there are several long-standing challenges that prevent ORR from implemented on commercial scale, including but not limited to the competition of four-electron pathway[6], the low solubility of $O_2$ in electrolytes[5], the decomposition of $H_2O_2$ product[5] and long-term system durability[7].

Another critical issue relatively unexplored is the impact of $O_2$ levels on the synthetic system. Since the first report of ORR a century ago[10], it seemed common conception that the alteration of $O_2$

[1]Key Laboratory for Soft Chemistry and Functional Materials, School of Chemistry and Chemical Engineering, Nanjing University of Science and Technology, Ministry of Education, Nanjing 210094, China. [2]Max Planck Institute of Colloids and Interfaces, Potsdam 214476, Germany. ✉e-mail: sheng.chen@njust.edu.cn

concentrations by diluents (like $N_2$) would disturb the equilibria of the reaction, and according to the Le Chatelier principle[11], resulting in performance losses.

Nevertheless, the exposure of a catalytic system to common diluents or impurities, especially under practically applied conditions, is almost unavoidable. An ORR electrochemical cell is composed of cathodic/anodic reactions separated by membranes and connected by electrolytes[5]. The crossover of gases/products derived from ORR or its coupled processes (like $N_2/CO_2$ from Fenton process[12,13], $C_2H_4$ from cascade synthesis[14] and $Cl_2$ from anodic seawater oxidation[15]) results in the high dilution of $O_2$. Ambient external gases may also directly transfer into the cathodic chamber to contact with ORR catalysts upon the membrane deactivation/device aging after long-term operation[3]. A promising strategy for $H_2O_2$ synthesis is to mimic natural enzyme superoxide dismutase, which generates $H_2O_2$ in full air atmosphere. The air composed of $O_2$ and $N_2$ is flushed into the electrochemical cells and dissolves near the reaction centers in minute amounts in the aqueous electrolyte. Under all the above scenario, the $O_2$ in mixed gas media quickly reacts away, and the decrease of $O_2$ levels usually reduces the initial reaction rates/equilibrium following the Le Chatelier principle[11], resulting in activity decay of $H_2O_2$ yields and Faradic efficiencies. The reaction is dependent on electrode substrate concentration and its diffusive replacement rate.

To the best of our knowledge, the majority of contributions still focus on the conversion of high-purity $O_2$ (>99.99%) prepared from air via the complex steps of separation, purification, compression and transportation, which would increase the production cost of $H_2O_2$. Developing a system that could achieve excellent ORR activities in a wide range of $O_2$ concentrations represent a significant progress toward practical applications. Importantly, this study also covers a typical problem of electrochemistry, i.e., it is difficult to exclude the diluents or impurities such as the crossover of gases/products in electrochemical systems. How is the partly complex dependence of the reaction rates on the gas partial pressure to be modeled and harnessed, so the system achieves improved activities?

The Eley-Rideal[16,17] and Langmuir-Hinshelwood mechanisms[18] are well-documented reaction models describing the adsorption and conversion at chemical/catalytic interfaces. The Eley-Rideal mechanism shows promise for ORR at high $O_2$ dilution because of only requires one of the reactant molecules to bind with the active site of catalysts followed by reacting directly with the other molecules from the volume phase. This is different from the Langmuir-Hinshelwood mechanism requiring both reactant molecules adsorbed onto the active sites of catalysts. The conventional Eley-Rideal mechanism only models the reactions at binary gas-solid interfaces. For gas-involving electrochemistry in liquids, like ORR, this model needs to be upgraded to include triple-phase interfaces, as high-rate ORR occurs at the ternary boundaries of $O_2$ gas/aqueous electrolytes/catalysts. Very recently, single-atom catalysts have been extensively reported in chemical/catalytic systems because of their excellent properties such as high metal utilizations and strong metal-support interactions[6,12]. While single-atom vacancy catalysts, the counterpart of single-atom catalysts except of using vacancies, have been still underexplored.

In this work, we find the reference zinc oxide (ZnO) crystal promoting ORR through Langmuir-Hinshelwood mechanism, which shifts to the Eley-Rideal mechanism after introducing single-zinc vacancies (denoted as Eley-Rideal-mechanism-ZnO or ER-ZnO). These vacancies allow for an enzyme-like binding of the catalyst with high selectivity to $O_2$ in mixed dioxygen media. The catalyst model shows practical no dependence of reaction rates on the $O_2$ partial pressure, and $O_2$ adsorption and availability are not the rate-determining step, even at low $O_2$ levels, i.e., the Le Chatelier's principle is out of importance in the system.

## Results

### Limitation of ZnO catalyst in mixed dioxygen electroreduction

The reference ZnO catalyst was synthesized by the controlled hydrolysis of zinc acetate (Supplementary Fig. 1), which was loaded onto gas-diffusion electrodes assembled in standard flow-type electrolytic cells for evaluating the ORR activities in mixed $O_2$ media (100%, 80%, 40%, and 21% $O_2$ in $O_2/N_2$ mixtures; Fig. 1a). The ORR activities were quantified by chronoamperometric tests at different current densities (Figs. 1b, c and Supplementary Fig. 2). Lowering $O_2$ concentrations from 100% to 21% with this reference ZnO catalyst, the $H_2O_2$ yields and Faradaic efficiencies show a significant decay at current densities between 50 and 300 mA cm$^{-2}$. More specifically at 300 mA cm$^{-2}$, the $H_2O_2$ yields are 25.6, 24.4, 17.6, and 14.4 mol $g_{cat}^{-1}$ h$^{-1}$ in 100%, 80%, 40%, and 21% $O_2$, respectively. This reflects the above-discussed oxygen depletion at the active centers, that is, the ORR turns transport limited at the triple-phase interfaces. Analogously, the $H_2O_2$ Faradaic efficiencies decrease from 91.5%, 87.3%, 63.0% to 51.5% for 100%, 80%, 40%, and 21% $O_2$, respectively (Fig. 1c). This result is further consistent to rotation ring disk electrode (RRDE) measurements in generator-collector mode (Fig. 1d–f), showing a declined $H_2O_2$ selectivity (from 70% to 49%) and elevated electron transfer numbers (from 2.6 to 3.2) with lowering $O_2$ concentrations from 100% to 21%, due to the lack of $O_2$ source and the wanted over-oxidization to water via four-electron-transfer pathway at the open binding sites. Note that we present the data on non-modified ZnO to illustrate the typical problems of ORR at mixed $O_2$ conditions and industrially relevant rates.

### Proposed solution and the characterization of ER-ZnO catalyst

Next, the ER-ZnO was synthesized via the same procedure as the reference ZnO except for adding glycerol. The glycerol was used as a modifier to synthesize zinc glycerate precursor[19]. During the calcination in air, a significant amount of O elements evaporated from zinc glycerate to form ZnO. Due to the strong interaction between Zn and O, some Zn elements also escaped from the material, resulting in structural Zn defects. The change in glycerol percentages can manipulate Zn defects in ER-ZnO. Figure 2a, b discloses the atomically dispersed zinc defects throughout ER-ZnO sample, which agrees well with the declined peak intensity at corresponding zinc-vacancy site (Fig. 2b, Supplementary Fig. 3), speaking the fact of some zinc atoms missing in the crystal structures (X-ray diffraction; Fig. 2c). The unobvious difference in XRD patterns between ER-ZnO and ZnO (i.e., only slight intensity alternations in (100), (002), and (101) crystal surfaces) provides additional evidence of the single-atom zinc vacancy nature. This conclusion is verified by other characterizations: the ER-ZnO exhibits amplified electron paramagnetic resonance (EPR) peak signal at $g = 2.0040$ relative to ZnO (4.53 vs. 0.31, Fig. 2d)[20]; the semiconductor structure of ER-ZnO also differs from the reference ZnO according to Mott-Schottky curves and electrochemical impedance spectroscopy (p-n type junction vs. n-type, Supplementary Figs. 4, 5)[21]. Quantitively, the zinc-vacancy concentration is determined to be 8.72% according to the Zn 2p/O 1s deconvolutions and overall survey of X-ray photoelectron spectra (XPS, Supplementary Fig. 6 and Supplementary Table 1).

The single-zinc vacancy nature of ER-ZnO is confirmed by X-ray absorption near edge structure (XANES, Fig. 2e) and extended X-ray absorption fine structure (EXAFS, Fig. 2f, g) analyses. ER-ZnO exhibits the same Zn K-edge XANES peak position as ZnO, which constitutes an upshift of 0.32 eV to high energy as compared to Zn foil (1.49 vs. 1.17 eV)[22]. This result agrees with the EXAFS of ER-ZnO and ZnO characteristic of two peaks assigning to Zn-O (1.97 Å) and Zn-Zn (3.26 Å) distances, sharply different from solely Zn-Zn distance (2.64 Å) for Zn foils (Fig. 2f, g, Supplementary Fig. 7). According to the EXAFS fitting curves (Fig. 2g–j; Supplementary Table 2), ER-ZnO has a Zn-Zn bond distance and coordination number greatly exceeding Zn foil (3.259 vs. 2.643 Å, 11.904 vs. 6.0)[23]. Interestingly, the increase in the Zn-O bond length (ER-ZnO:

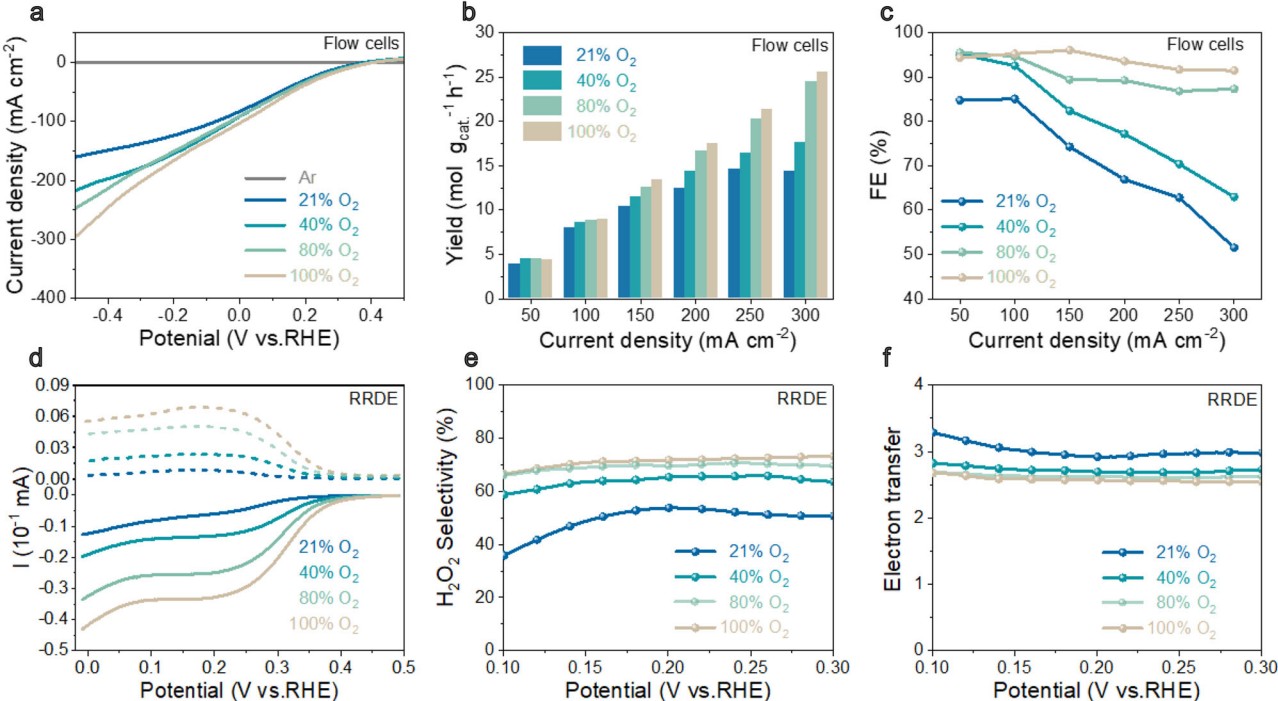

**Fig. 1 | The limitation of reference ZnO catalyst for oxygen electroreduction to $H_2O_2$ in mixed $O_2$ media (21%, 40%, 80%, and 100% $O_2$ in $O_2/N_2$ mixtures). a** LSV curves tested in flow cells. **b** $H_2O_2$ yield s tested in flow cells. **c** $H_2O_2$ Faradaic efficiencies tested in flow cells. **d** LSV polarization curves on RRDE at a rotation speed of 1600 rpm and scan rate of 5 mV s⁻¹, where the $H_2O_2$ currents were recorded on the ring electrode at a constant potential of 1.2 V (vs. RHE). **e** $H_2O_2$ selectivity calculated on the basis of RRDE. **f** Electron transfer number ($n$) calculated on the basis of RRDE.

1.968 Å vs. ZnO: 1.956 Å) is caused by the local deletion of Zn atoms, which migrates nearby O atoms to the defects and increases the Zn-O bond length. This is consistent with the increase in Zn−Zn bond length due to the missing of local Zn atoms (ER-ZnO: 3.259 Å vs. ZnO:3.234 Å) and decrease in the coordination number of Zn-O (ER-ZnO: 3.906 vs. ZnO: 4.006) and Zn-Zn (ER-ZnO: 11.904 vs. ZnO: 12.083) due to the unsaturated Zn sites inside the material[22,24]. Notably, our experimental results show the absence of O defects, otherwise Zn−Zn distance around the O defects would be shortened (as also confirmed by XPS elemental analysis in Supplementary Fig. 6 and EPR in Fig. 2d).

In the O K-edge XANES of ER-ZnO and ZnO (Supplementary Fig. 8), the pre-edge peaks (535.1 and 537.9 eV) are attributed to the unoccupied hybridized states of O 1$s$ electrons transitioning to Zn 3$d$ and O 2$p$ orbitals above Fermi energy levels, splitting into two asymmetric peaks of different energies of $t_{2g}$ and $e_g$. The broad peak represents the electron transition of O 1$s$ to hybridized orbitals of O 2$p$ and Zn 4$sp$ states, while the sharp peaks represent the electronic transitions of O 1$s$ to the more localized O 2$p_z$ and O 2$p_{x+y}$ states[25]. Notably, the peak intensity of ER-ZnO (at around 535.1–537.9 eV) slightly decreases due to the reduction of available empty O 2$p$ states. This suggests more charge transfer from Zn to O atoms, resulting in an increase in Zn valence state and a decrease in O valence state. These findings are consistent with Zn K-edge XANES data. Further, with the elevated oxidation state of Zn, the number of outer electrons in Zn atoms decreases, which can facilitate the bonding with electron-rich O atoms and contribute to the selective adsorption of $O_2$ by ER-ZnO in the mixed $N_2/O_2$ atmosphere. At the same time, due to the decrease in outer electrons, it is difficult to provide additional electrons for bonding with O atoms, which has resulted in smaller adsorption strength of $O_2$ on the ER-ZnO as compared to ZnO, endowing ER-ZnO with improved two-electron ORR activity.

We find the electronic structure closely related to electrochemical activities. ER-ZnO shows the increased Zn oxidation valence state and decreased number of outer electrons of the Zn atoms that facilitate

bonding with electron-rich oxygen atoms. This can promote the selective adsorption of $O_2$ by ER-ZnO in a mixed $N_2/O_2$ atmosphere. Further, due to the decrease of outer electrons, ER-ZnO cannot provide sufficient electrons for bonding with O atoms, resulting in appropriate adsorption strength of *$O_2$ as compared to ZnO. According to the Sabatier's principle[26], the moderate adsorption strength for reactive species on ER-ZnO can demonstrate improved two-electron ORR activity[27]. These represent for us the possibility for an exciting new modulation of the atomic structure and the related electronic properties for ER-ZnO, as we can assume the presence of more electrophilic sites that allow for improved, selective $O_2$ binding and thereby enhanced ORR activities.

## Quantification of performances in mixed dioxygen electroreduction

Consequently, the electrocatalytic ORR performances of ER-ZnO were tested in the same condition as the reference ZnO (Supplementary Figs. 9–12). The linear sweep voltammetry (LSV) was conducted for ER-ZnO-based electrodes in Ar- and $O_2$-saturated electrolytes, where the current densities in all $O_2$-saturated electrolytes surpass that in Ar-counterpart indicating the occurrence of oxygen electroreduction (Fig. 3a)[28]. The ORR activities were quantified by chronoamperometric tests (Fig. 3b, c). Lowering $O_2$ concentrations from 100% to 21% with the ER-ZnO catalyst, the $H_2O_2$ yields and Faradaic efficiencies show minute decay at current densities between 50 and 300 mA cm⁻². More specifically at 300 mA cm⁻², the $H_2O_2$ yields are 25.89, 25.60, 25.03, and 24.99 mol g$_{cat}$⁻¹ h⁻¹ in 100%, 80%, 40%, and 21% $O_2$, respectively. Analogously, the $H_2O_2$ Faradaic efficiencies are 92.5%, 91.5%, 89.5% to 89.3% for 100%, 80%, 40%, and 21% $O_2$, respectively (Fig. 3c). Therefore, the accessibility to $O_2$ is not the rate-limiting step anymore. In other words, we move the Le Chatelier reaction kinetics out of importance in the catalytic system.

The outstanding performances of ER-ZnO are validated by RRDE on drop-casting electrodes in generator-collector mode (Fig. 3d–f and

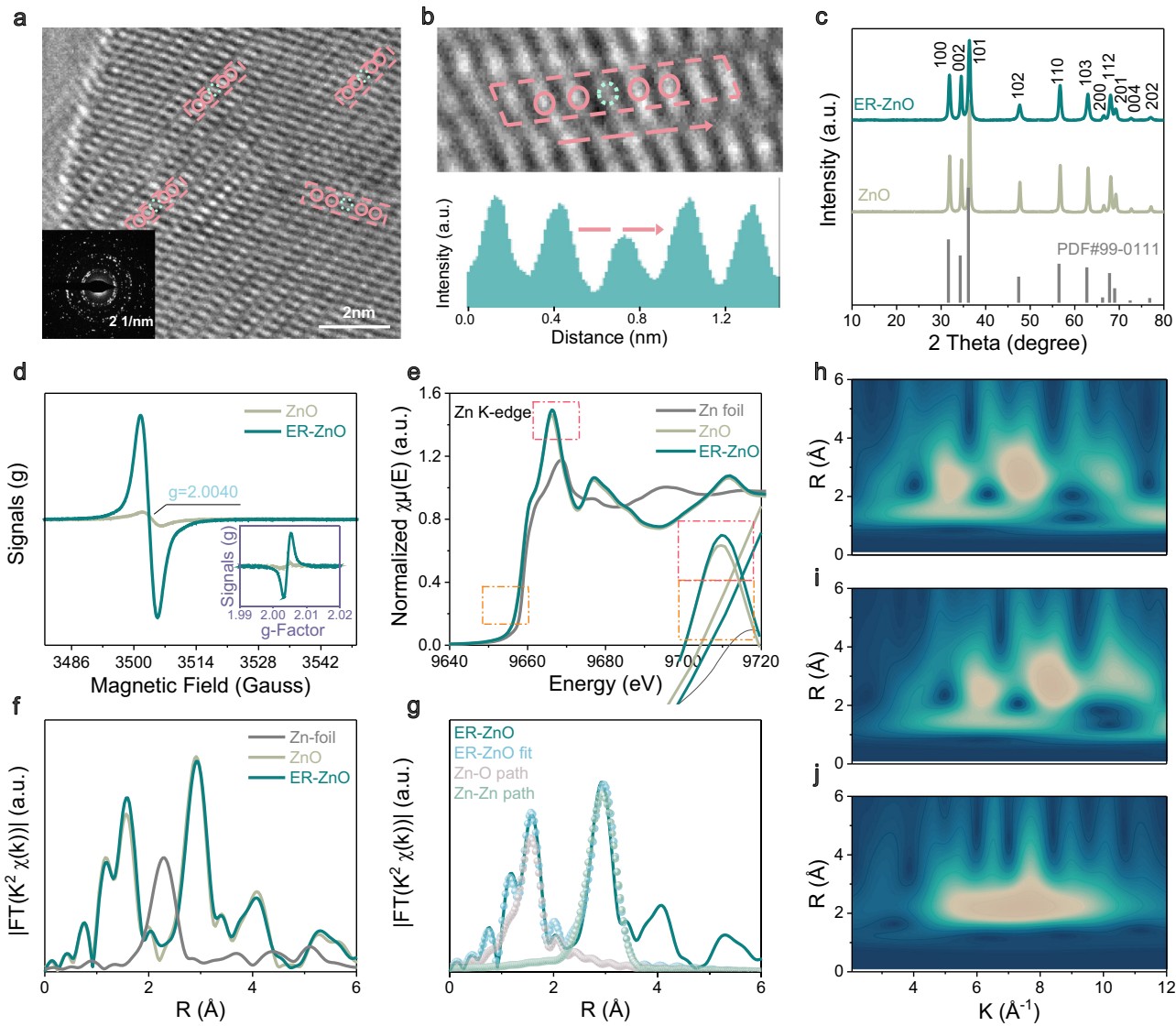

**Fig. 2 | The structural characterizations of ER-ZnO catalyst. a, b** High-resolution transmission electron microscopy (HR-TEM) images with typical lattice defects marked in red lines, with the intensity profile recorded for the selected area. **c** X-ray diffraction (XRD) patterns of ER-ZnO and ZnO. **d** The electron paramagnetic resonance (EPR) spectra of ER-ZnO and ZnO, with the inset as the data reproduced by g-Factor abscissa. **e** X-ray absorption near edge structure (XANES) spectra of ER-ZnO, ZnO, and Zn foils. **f** Extended X-ray absorption fine structure (EXAFS) spectra of ER-ZnO, ZnO and Zn foils. **g** Fitting curves of EXAFS spectra of ER-ZnO, ZnO and Zn foils. **h–j** The corresponding wavelet transform (WT)-EXAFS contour plots of ER-ZnO, ZnO and Zn foils.

Supplementary Figs. 13–16). The RRDE activities were calculated on the basis of disk current ($I_d$) at the spin rate of 1600 rpm and ring current ($I_r$) at the constant potential of 1.2 V. ER-ZnO, demonstrating good $H_2O_2$ molar fraction selectivity (70–90%, Fig. 3e) with the transferred electron numbers (2.0–2.6) close to that of theoretical two-electron-transfer pathway (Fig. 3f). Importantly, the ER-ZnO has demonstrated longstanding and stable $H_2O_2$ production in both mixed and high-purity $O_2$ media (21% $O_2$ in Fig. 3g, h and 100% $O_2$ in Supplementary Fig. 17, Supplementary Table 4). Even under the industrial-level current densities, the ER-ZnO catalyst tested in 0.6 M $K_2SO_4$ has shown seldom activity degradation for 100 h at 200 mA cm$^{-2}$. In ten-times repetitive chronoamperometric cycles, the acquired $H_2O_2$ concentration can accumulate to 350 mg L$^{-1}$ in 1 L of 0.6 M $K_2SO_4$ every 3 h, with the Faradaic efficiencies above 90%.

**Revealing the mechanism of mixed dioxygen electroreduction**

Motivated by the excellent ORR activities, experimental and theoretical studies are conducted to understand the catalytic mechanisms. Figure 4a, b shows the operando Raman spectra for capturing the short-lifetime ORR adsorbates (21% $O_2$ in $O_2/N_2$ mixtures, Supplementary Figs. 18–20). A flow-type cell with a quartz window was developed to detect the Raman signal at 532 nm excitation wavelength from 0 to −0.8 V in 200 mV steps. The spectra of both ER-ZnO and ZnO display a subset of the following vibrational data: the band at 978 cm$^{-1}$ originated from $SO_4^{2-}$ from the electrolytes[29]; the band in 1350–1600 cm$^{-1}$ from carbon paper substrate; two potential-dependent bands at 525 and 845 cm$^{-1}$ from O–O stretching mode of surface bound *$O_2$ and *$OOH$, respectively[30]. The peak intensities of *$O_2$ and *$OOH$ have been analyzed in Fig. 4c. The changes in surface intermediates show the characteristic peaks for *$O_2$ and *$OOH$ in response to applied potentials. For L-H mechanism, the proton occupies part of active sites, resulting in a relatively low concentration of *$O_2$ on the surface and consequently a low peak intensity (for reference ZnO). In contrast, the E-R mechanism allows for more active sites for the adsorption of $O_2$, resulting in a higher *$O_2$ peak intensity (for ER-ZnO). Further examination of ER-ZnO reveals the tendency of *$O_2$ peak intensity decrease while *$OOH$ increases with elevated applied potentials. This originated from E-R mechanism that causes the direct

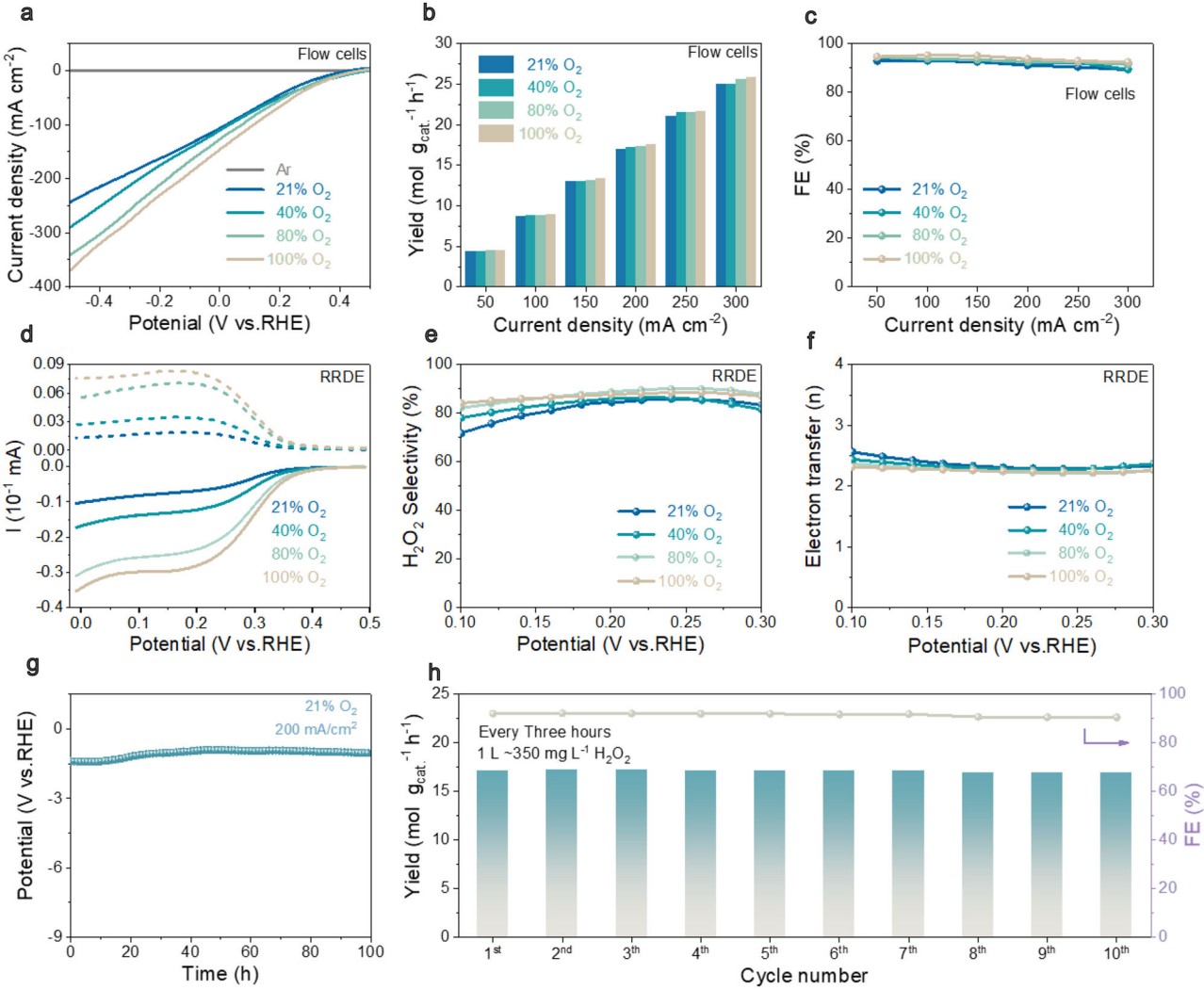

**Fig. 3 | The advantages of ER-ZnO catalyst for oxygen electroreduction to $H_2O_2$ in mixed $O_2$ media (21%, 40%, 80%, and 100% $O_2$ in $O_2/N_2$ mixtures). a** LSV curves tested in flow cells. **b** $H_2O_2$ yields tested in flow cells. **c** $H_2O_2$ Faradaic efficiencies tested in flow cells. **d** LSV polarization curves on RRDE at a rotation speed of 1600 rpm and scan rate of 5 mV s$^{-1}$, where the $H_2O_2$ currents were recorded on the ring electrode at a constant potential of 1.2 V (vs. RHE). **e** $H_2O_2$ selectivity calculated on the basis of RRDE. **f** Electron transfer number ($n$) calculated on the basis of RRDE. **g** The durability test of ER-ZnO in 21% $O_2$ with the current density of 200 mA cm$^{-2}$. **h** The $H_2O_2$ yields and Faradaic efficiencies of the chronopotentiometry test under 200 mA cm$^{-2}$ in ten times 3-h cycle tests.

coupling between $*O_2$ and dissociative H in the electrolyte, resulting in a rapid reaction rate than $*OOH$ protonation to produce $H_2O_2$. Due to the sufficient supply of $*O_2$ species in E-R mechanism, $*OOH$ continuously accumulates on the surface of ER-ZnO, leading to the simultaneous decrease of $*O_2$ species. On the other hand, the change of $*OOH$ peak intensity shows the similar tendency to $*O_2$ peak intensity on the ZnO surface. This originated from L-H mechanism, where $*O_2$ binds to $*H$ before generating $*OOH$, resulting in a parallel change of these two intermediates.

To experimentally probe their $O_2$ selectivity, temperature programmed desorption (TPD) was conducted for the catalysts from 50 to 800 °C at the heating rate of 5 °C min$^{-1}$ (Supplementary Fig. 21). The $O_2$-TPD curves of both ER-ZnO and ZnO disclose three peaks at 404.9, 528.5, and 772.7 °C assigning to surface adsorbed $O_2$, suboxide formation and removal lattice oxygen, respectively (Supplementary Fig. 21a). The ER-ZnO has overtaken ZnO by the integrated peak area for surface adsorbed $O_2$ (0.103 vs. 0.102), reflecting its high selectivity to $O_2$ by providing more active centers for absorbing oxygen species. This result agrees with $N_2$-TPD curves of ER-ZnO containing of two bands at 412.3 (surface adsorbed $N_2$) and 528.1 °C (lattice nitrogen, Supplementary Fig. 21b), demonstrating less tightly binding with $N_2$ by

showing smaller integrated peak areas for surface adsorbed $N_2$ than ZnO counterpart (0.104 vs. 0.167). By calculating the peak area ratios of $O_2$–TPD/$N_2$–TPD curves, the chemical $O_2$ selectivity of ER-ZnO relative to the reference ZnO is 1/0.611 (Supplementary Fig. 21c).

To elucidate the underlying origin of high $O_2$ selectivity, density function theory (DFT) calculations were performed for ER-ZnO and ZnO in the mixed media (Supplementary Figs. 22–28). The density of states of Zn 3d orbital show significant occupied states near the Fermi level for both ER-ZnO and ZnO (Supplementary Fig. 29), elaborating their possible bindings with the absorbates in mixed $O_2/N_2$ electrochemical systems, including $*O_2$ from $O_2$, $*OOH$, $*N_2$ from $N_2$ and $*O_2$-$*H$ from the protons via the dissociation of aqueous electrolytes (Supplementary Figs. 30–33)[31]. The adsorption energy levels on ER-ZnO are −1.00 eV for $*O_2$, −1.51 eV for $*OOH$, −0.34 eV for $*N_2$ and 0.59 eV for $*O_2$-$*H$ (positive value indicative of unstable adsorption), and in accordance the reference ZnO are −3.77 eV for $*O_2$, −2.62 eV for $*OOH$, −1.25 eV for $*N_2$, and −3.63 eV for $*O_2$-$*H$, respectively (Fig. 4d). Different from the reference ZnO favorably adsorbing all the species of $O_2$, $*OOH$, $N_2$ and protons, the ER-ZnO only binds stably with $O_2$ and $*OOH$. Appropriate $*OOH$ adsorption indicates it is stable on the surface, leading to high reaction activity

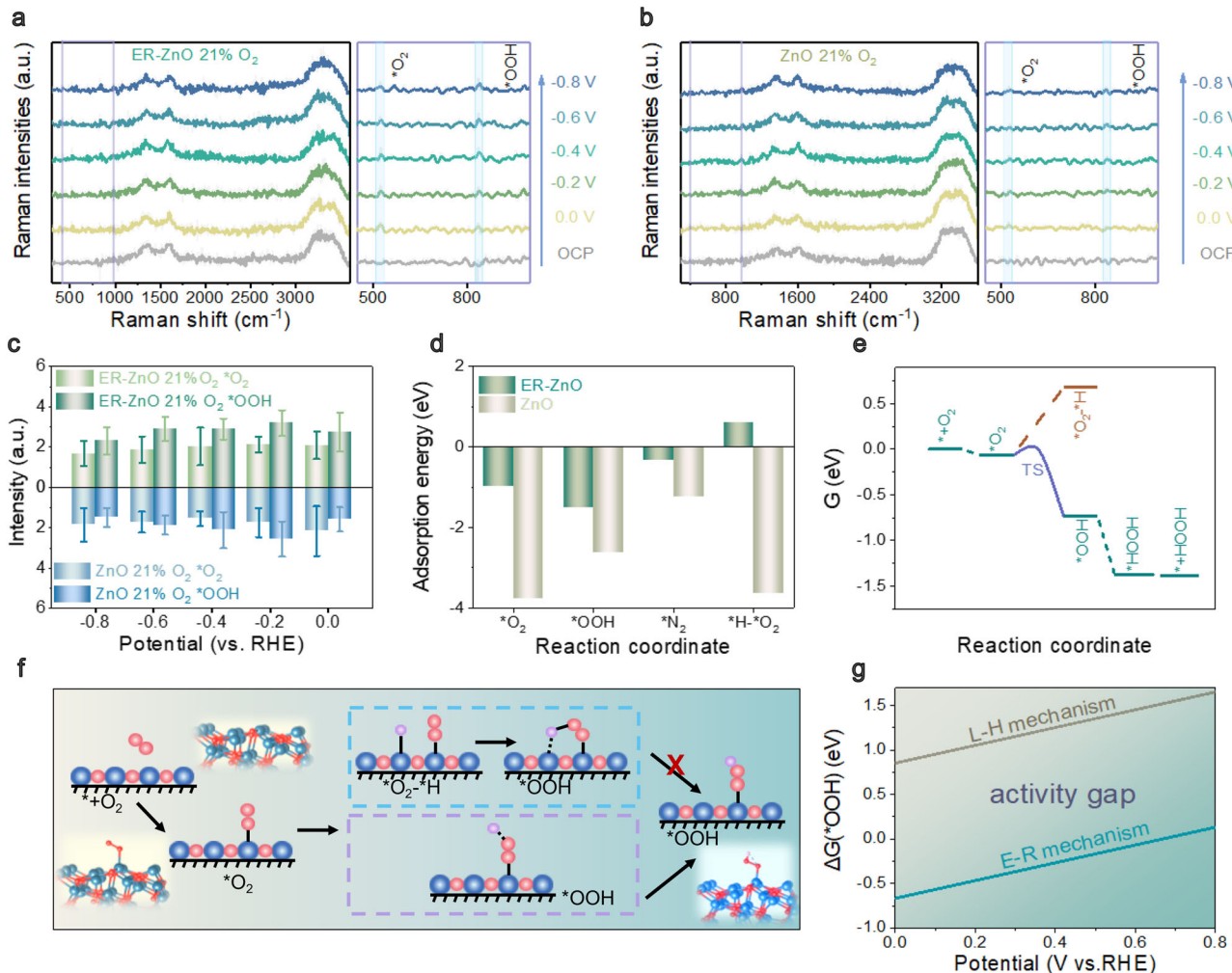

**Fig. 4 | Mechanism investigation in mixed $O_2/N_2$ media (i.e., 21%$O_2$−air). a** The operando Raman spectra of ER-ZnO. **b** The operando Raman spectra of the reference ZnO. **c** The peak intensity analyses with error bar in operando Raman test. Five measurements were conducted for each data point with the error bars corresponding to the standard deviation. **d** The adsorption energy of different reaction intermediates. **e** Density functional theory (DFT) calculated free energy diagrams of ORR on ER-ZnO. **f** Reaction pathways of Eley-Rideal mechanism and Langmuir-Hinshelwood mechanism for ER-ZnO. **g** Kinetic barriers for the hydrogenation of *$O_2$ to *OOH on ER-ZnO via Eley-Rideal and Langmuir-Hinshelwood mechanisms.

for two-electron ORR. Further, ER-ZnO shows high selectivity to $O_2$ by displaying larger binding energy relative to $N_2$ (−1.00 vs. 0.34 eV), which is necessary for ORR proceeding in mixed $O_2/N_2$ atmosphere. Next, the intermediate *$O_2$-*H plays a crucial role in the L-H mechanism. Upon analyzing the adsorption energies, the adsorption energy of *$O_2$-*H is positive (0.59 eV), indicating the stabilization of *$O_2$-*H intermediate in L-H mechanism is challenging on the surface of ER-ZnO. This is different from the negative adsorption energy on reference ZnO surface with L-H mechanism (*$O_2$-*H adsorption energy: −3.63 eV). Therefore, ER-ZnO prefers E-R mechanism while ZnO prefers L-H mechanism.

The above result is consistent with the difference in charge densities (Supplementary Figs. 34, 35), which displays the favorable adsorption of ER-ZnO with *$O_2$ and *$N_2$ characteristic of dense accumulated electron clouds relative to *$O_2$-*H[32]. Quantitatively, the electron transfer numbers between ER-ZnO and different absorbates (*$O_2$, *$N_2$ and *$O_2$-*H) have been determined by Bader charge transfer analyses (Supplementary Table 5). ER-ZnO prefers to bind with *$O_2$ by transferring more electrons relative to *$N_2$ (0.05 vs. 0.02e). Different from the reference ZnO characteristic of significant charge migrations after absorbing all the absorbates (0.6e for *$O_2$, 0.18e for *$N_2$ and 0.67e/0.42e for *$O_2$/*H of *$O_2$-*H), ER-ZnO could only bind with *$O_2$ and *$N_2$

rather than *$O_2$-*H (0.05 e for *$O_2$, 0.02 e for *$N_2$, and −0.05e/−0.27e for *$O_2$/*H of *$O_2$-*H).

Overall, the reaction pathways of ORR are proposed for ER-ZnO and ZnO by using Gibbs free energy as a descriptor (Figs. 4e, f). Two-electron-transfer ORR commonly consists of four cascade steps of the catalysts adsorbing/activating dioxygen to form *$O_2$, two consecutive protonations to *OOH and *HOOH, and the dissociation to $H_2O_2$ for recovering the active centers. Particularly in the potential-limiting step of *$O_2$ → *OOH, two fundamental mechanisms might occur according to their proton sources (Fig. 4f, Supplementary Fig. 36): i) directly stem from bulk phase of aqueous electrolytes via E-R mechanism; ii) from the adjacent adsorbed *H via L-H mechanism. As revealed in Fig. 4e–g, the ER-ZnO prefers E-R mechanism with a free energy change of only −0.67 eV as compared to L-H mechanism (0.75 eV). This result is supported by the transition state computations of E-R mechanism for ER-ZnO by a climbing image nudged elastic band method with implicit solvation model (Supplementary Fig. 37), showing excellent reaction kinetics with the transition energy gap of only 0.057 eV (Supplementary Fig. 37). In great contrast, the reference ZnO counterpart tends to proceed via L-H mechanism rather than E-R mechanism under the same computation condition (free energy change of 0.37 vs. 1.28 eV, Supplementary Figs. 38–41).

## Prototype electrolysis device and techno-economic analysis

To demonstrate the feasibility of the present model system for on-site $H_2O_2$ production, a prototype device was assembled (Fig. 5a–c), which is an integrated standalone box (size: 15 cm × 15 cm × 19 cm) composed of rechargeable battery as power sources, gas (air) pumps for flowing gases, peristaltic pumps for circulating electrolytes and flow-type cells loaded with ER-ZnO catalysts. Their ORR activities were quantified by chronoamperometric tests at different current densities (Supplementary Fig. 42). Lowering $O_2$ concentrations from 100% to 21% in mixed dioxygen gas, the ORR activities of the prototype device display seldom activity decay in the current density range of 50–300 mA cm$^{-2}$. At the current density of 300 mA cm$^{-2}$, the prototype device demonstrates $H_2O_2$ yields (25.78, 25.94, 25.63, and 24.39 mol g$_{cat}^{-1}$ h$^{-1}$) and Faradaic efficiencies (92.15%, 92.71%, 91.60%, and 87.17%) that comparable to above flow-type electrochemical configurations in 100%, 80%, 40%, and 21% $O_2$, respectively. Even in much lower $O_2$ concentrations (i.e., 5%, 10% and 15% $O_2$; Supplementary Figs. 43–45), the

prototype device can still show superior activities approaching theoretical limits (Supplementary Fig. 46).

Next, the prototype device has operated directly using atmospheric air feedstock, displaying the $H_2O_2$ yields and Faradaic efficiencies (such as 24.41 mol g$_{cat}^{-1}$ h$^{-1}$ and 89.24% at 300 mA cm$^{-2}$) comparable to those in mixed dioxygen media. This result is consistent to RRDE test showing analogous results to mixed dioxygen (Supplementary Fig. 47). By elevating the current density to 320 mA cm$^{-2}$, the prototype device has demonstrated the highest $H_2O_2$ yield and Faradaic efficiency (26.2 mol g$_{cat}^{-1}$ h$^{-1}$ and 87.76%; Fig. 5c and Supplementary Fig. 48), approaching the theoretical limit for direct air electrolysis (-345.8 mA cm$^{-2}$). To our best knowledge, the ORR activity of ER-ZnO rates as the best in the literature for direct air electrolysis (Fig. 5d and Supplementary Table 6).

Consequently, the practical application of the prototype device was demonstrated for the in situ oxidative degradation of a model dye used for referencing degradation experiments[33]. To evaluate the degradation

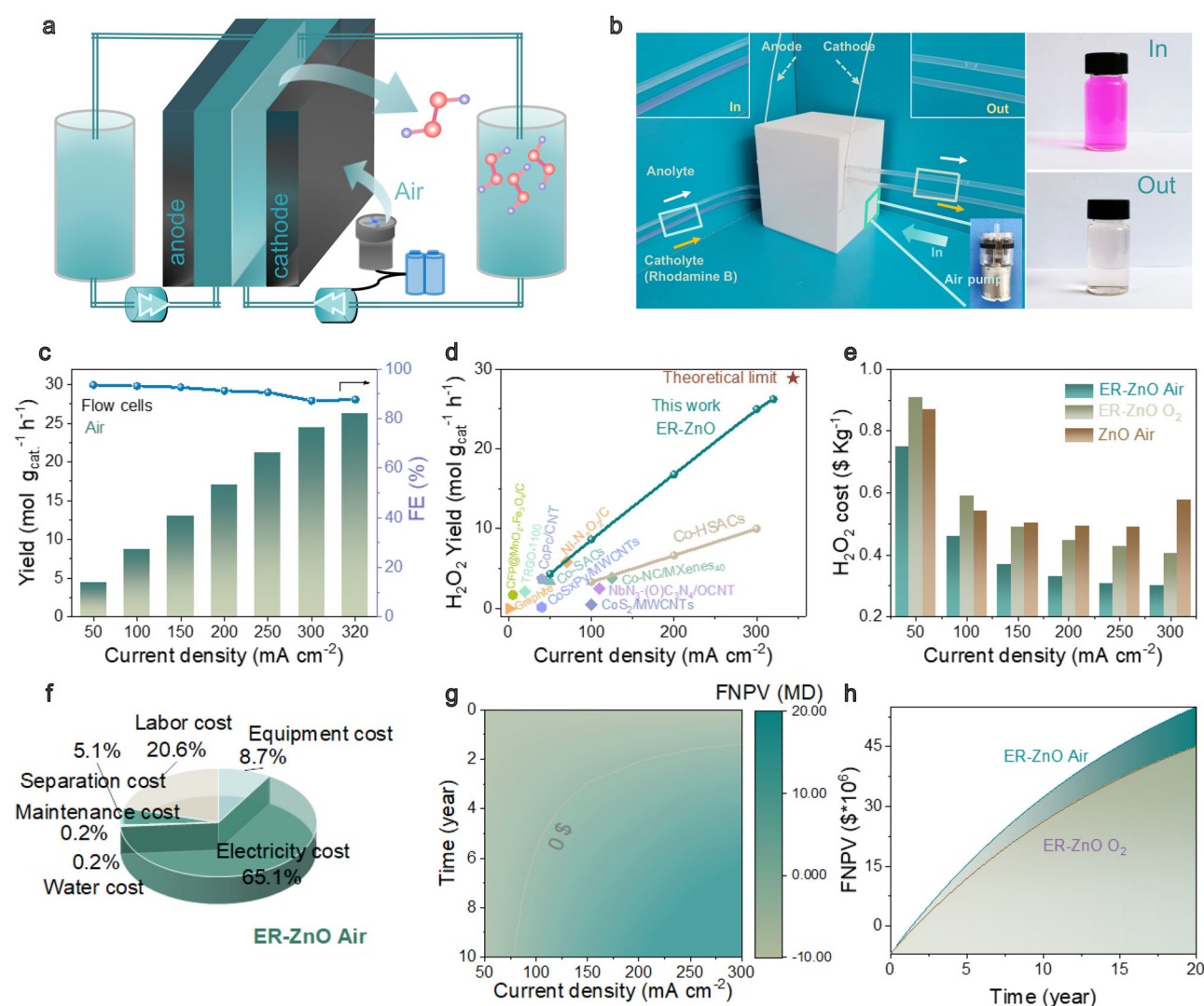

**Fig. 5 | Prototype device and techno-economic analyses for oxygen electro-reduction using atmospheric air feedstock. a** The schematic diagram of the prototype device. **b** The optical picture of the prototype device operating for pollutant degradation of rhodamine B (The background color has changed from blue to green for reliable comparison). **c** The ORR $H_2O_2$ yield rates of ER-ZnO catalyst in the prototype device in atmospheric air. **d** The comparison of $H_2O_2$ yields with the state-of-the-art literature in atmospheric air, as listed in

Supplementary Table 4. **e** Techno-economic analyses showing the $H_2O_2$ production cost for ER-ZnO and ZnO catalysts in atmospheric air and 100% $O_2$ (pure dioxygen). **f** The proportions of $H_2O_2$ production cost for ER-ZnO in atmospheric air. **g** Financial net present value (FNPV) analyses of $H_2O_2$ production for ER-ZnO catalyst in atmospheric air. **h** Comparison of FNPV of $H_2O_2$ production for ER-ZnO in atmospheric air and 100% $O_2$ (pure dioxygen).

ability, an aqueous solution was made of rhodamine B in $K_2SO_4$ aqueous electrolyte. The solution flowed through the prototype device operating in atmospheric air at 300 mA cm$^{-2}$, where the color of rhodamine B solution fades rapidly. The optical images of electrolyte (Fig. 5b) and Supplementary video clearly displayed the dye decolorization after the prototype device operating in the established condition.

Finally, techno-economic analyses have been conducted under scalable and industrially viable conditions[33,34]. The $H_2O_2$ production cost from ORR contains of capital investment and operating cost, linking to more detailed system set-ups of energy (electricity), feed-stocks (mixed $O_2$ source and water), electrolysis cell, labor cost, maintenance cost, etc. Different from many previous reports of roughly calculating the average $H_2O_2$ cost from only energy and feedstock input, here a rigorous simulation model has been built by taking account of all of the possible parameters (Supplementary Tables 7–9). The accuracy of the models was examined by comparing the simulation results of ER-ZnO at 300 mA cm$^{-2}$ with that from manual calculations according to Faraday law, and both of the data are consistent (please see experimental section for details).

The cost breakdown for $H_2O_2$ is calculated by normalizing both capital investment and operating cost, and presented as a function of current densities (Fig. 5e). In the mixed $O_2$ condition (i.e., 21% $O_2$ or air), the $H_2O_2$ production cost of ER-ZnO decreases with the elevated current densities, from $0.75 kg$^{-1}$ at the current density of 50 mA cm$^{-2}$ sharply down to $0.33 kg$^{-1}$ at 200 A cm$^{-2}$, and slowly to the lowest price of $0.30 kg$^{-1}$ at 300 mA cm$^{-2}$. While under the same test condition in pure dioxygen (i.e., 100% $O_2$), the minimal $H_2O_2$ production costs for ER-ZnO are 0.40 $ kg$^{-1}$ at 300 mA cm$^{-2}$, indicating the merit of mixed $O_2$ sources for economic manufacture. Notably, the $H_2O_2$ production costs of ER-ZnO ($0.30 kg$^{-1}$) only account 20% of market prices from traditional anthraquinone oxidation/reduction process ($1.5 kg$^{-1}$).

To examine the contributions of different parameters to $H_2O_2$ production cost, single variable sensitivity has been analyzed for ER-ZnO (Fig. 5f, Supplementary Fig. 49 and Tables 10, 11). In the mixed $O_2$ condition (21% $O_2$-air), the $H_2O_2$ production cost is most susceptible to the variation in electricity price (65.1%), followed by human labor (20.6%), equipment (8.7%), separation (5.1%), maintenance (0.2%) and water cost (0.2%). Therefore, cheap electricity is critical to achieving low production cost, that is, with each US$0.01 kWh$^{-1}$ corresponding to a $H_2O_2$ price change of US$0.002 kg$^{-1}$ (Fig. 5e). This result is different from the $H_2O_2$ production cost of ER-ZnO in pure dioxygen (100% $O_2$, Supplementary Fig. 49), showing it susceptible to the variation in both electricity (51.3.1%) and $O_2$ (24.2%) prices, followed by human labor (14.5%), equipment (6.1%), separation (3.6%), water cost (0.2%) and maintenance (0.1%). Each US$0.01 kWh$^{-1}$ of $O_2$ corresponds to a $H_2O_2$ price change of US$0.001 kg$^{-1}$ (Supplementary Fig. 49).

The accumulative economic profit has been evaluated by financial net present value (FNPV) according to the following equation:

$$FNPV = \sum_{t=0}^{n} (Cl - CO)_t \times (1+i)^{-t} \qquad (3)$$

where $Cl$ is the present value of future cash flow ($H_2O_2$ product revenue), $CO$ is the present value of the original investment (capital cost, operating cost and 25% tax), $i$ and $t$ are the discount rates and the duration (payback time), respectively. The FNPV and payback time for ER-ZnO in mixed $O_2$ media (21% $O_2$ or air) are illustrated according to the current densities (Fig. 5g and Supplementary Fig. 42). At the operating current density of 50 mA cm$^{-2}$, the FNPV for ER-ZnO starts at $-6,948,946 and ends at $-5,107,495 through the whole lifespan. The negative FNPV values indicate a non-profit manufacture. Notably, the payback duration is reduced by elevating current densities, for example, 6 years for operating at 100 mA cm$^{-2}$ (FNPV value: $8,606,199) and 2 years for operating at 300 A cm$^{-2}$ ($54,815,167),

underlining the importance of high reaction rates for economic manufacture. Besides current densities, the $O_2$ source is underlined as an important evaluation parameter, because ER-ZnO in pure dioxygen (100% $O_2$) only arrives at $45,048,833 at 300 mA cm$^{-2}$ under the same condition (Fig. 5h and Supplementary Tables 10, 11). The use of a mixed $O_2$ source (21% $O_2$ or air) could bring 21.7% better economics for ER-ZnO as comparison to that in purity dioxygen (100% $O_2$). Collectively, the techno-economic analyses indicate our model is a promising candidate for industrial $H_2O_2$ synthesis.

## Discussion

The recent studies show tremendous examples of how to harness chemical/catalytic systems by leveraging specific reaction parameters (such as concentrations[35], pressures[36] and temperatures[37]), presumably under the assumption of the basic rule of Le Chatelier principle applying. In this work, we have developed a single-atom vacancy model for the oxygen electroreduction to $H_2O_2$, which apparently bypasses the macroscopic Le Chatelier's kinetics. We show the prominent ORR activities in a wide range of $O_2$ levels. Further techno-economic analyses demonstrate the advantages of oxygen electroreduction in mixed dioxygen condition as comparison to high-purity $O_2$ and conventional anthraquinone process. This work would have general implications, giving insight into the rational design and tailoring of heterogeneous catalytic systems "beyond" the constraint of classical Le Chatelier principle, approaching the substrate binding behavior typical for natural enzymes.

## Methods

### Material synthesis

ER-ZnO was synthesized by dissolving $Zn(OAc)_2$ (0.26 g) in a mixed solution containing of ethanol (30 mL) and glycerol (2 mL). Next, the mixture was transferred into Teflon-lined stainless-steel autoclave for hydrothermal reaction at 180 °C for 24 h. The as-obtained solid precursor (i.e., zinc glycerolate) was collected and thermally annealed at 400 °C for 1 h. Other comparison samples, such as ZnO and ER-ZnO-X ($X$ = 1, 3 and 4), were synthesized via a similar procedure without glycerol or with different amounts of glycerol (i.e., 1, 3, and 4 mL, respectively).

### Rotating ring disk electrode (RRDE) system

Pt wire and Ag/AgCl electrodes were used as counter and reference electrodes, respectively. RRDE (PINE Research Instrument) was employed as working electrode. The catalyst ink was prepared by dip-coating method as follows: 5.0 mg of catalyst powder, 1.0 mg of carbon black and 30 μL of Nafion solution were dispersed in 970 μL of iso-propanol. After bath sonication for 30 min, 10 μL of the as-obtained catalyst ink was dropped onto the surface of disk electrode of RRDE followed by dry under ambient condition (catalyst loading: 0.2 mg cm$^{-2}$).

All of the ORR data were measured in 0.6 M $K_2SO_4$ aqueous solution. Prior to the test, cycle voltammetry measurement was applied on RRDE at the scan rate of 50 mV s$^{-1}$ until reaching a stable state. Then the ORR polarization curves were obtained by LSV with a sweep rate of 10 mV s$^{-1}$ at 1600 rpm. To detect the as-generated $H_2O_2$, the potential of 1.2 V (vs. RHE) was applied to the Pt ring electrode. The $H_2O_2$ selectivity and electron transfer number ($n$) were calculated on the basis of disk current ($I_d$) and ring current ($I_r$) according to the following equations:

$$H_2O_2\ selectivity\ (\%) = 200 \frac{I_r/N_c}{|I_d| + I_r/N_c} \qquad (4)$$

$$n = 4 \frac{|I_d|}{|I_d| + I_r/N_c} \qquad (5)$$

## Flow-type electrochemical cells

The flow-type cell was built by anodic and cathodic compartments separated by Nafion 115 membrane. The ER-ZnO catalyst was loaded on the GDE (working area: 1 cm$^2$) with a mass loading of 0.2 mg cm$^{-2}$. Next, the GDE attached with Teflon film was used as the cathode. IrO$_2$-coating titanium sheet and Ag/AgCl electrode were employed as the anode and reference electrode, respectively. Particularly for cathodic and anodic compartments, 50 mL 0.6 M K$_2$SO$_4$ aqueous solution was used as the electrolyte and recycled with the peristaltic pump at the rate of 40 r min$^{-1}$. The H$_2$O$_2$ concentration was measured after operatingg for 5 mins. The gas supply rate was stabilized at 20 mL min$^{-1}$ feeding into the cathodic compartment.

## Determination of H$_2$O$_2$ product

The H$_2$O$_2$ concentration was detected by using UV−vis spectra with TiOSO$_4$ chromogenic reagent. Specifically, 9 mL of cathodic electrolyte was mixed with 1 mL of TiOSO$_4$ chromogenic reagent composed of titanium (IV) sulfate (14 mg) and concentrated sulfuric acid (0.2 mL). The diluted sulfuric acid can hydrolyze titanium sulfate into titanium oxysulfate (TiOSO$_4$), and react with H$_2$O$_2$ in the cathodic electrolyte to form a yellow complex according to the following equations:

$$Ti(SO_4)_2 + H_2O = TiOSO_4 + H_2SO_4 \qquad (6)$$

$$TiOSO_4 + H_2O_2 = [TiO(H_2O_2)]SO_4 (Yellow\ complex) \qquad (7)$$

Subsequently, the amount of H$_2$O$_2$ was detected by analyzing the homogeneous mixed solution through UV−vis spectroscopy at the wavelength of 408 nm (supplementary Fig. 2).

The H$_2$O$_2$ yield (mol g$^{-1}_{cat}$ h$^{-1}$) and Faradaic efficiency (FE) for the H$_2$O$_2$ production were calculated according to the equation:

$$Yield = \frac{C_{H_2O_2} V_{electrolyte}}{t\ m_{cat}} \qquad (8)$$

$$FE\ (\%) = \frac{2F C_{H_2O_2} V_{electrolyte}}{34 It} \times 100\% \qquad (9)$$

Where $C_{H_2O_2}$ is H$_2$O$_2$ concentration (mol L$^{-1}$), $V_{electrolyte}$ is the volume of cathodic electrolyte ($L$), $F$ is the Faraday constant (96485 C mol$^{-1}$), 34 is the molar mass of H$_2$O$_2$, $t$ is reaction duration, $I$ is the applied steady current and $m_{cat}$ is catalyst mass loading (mg cm$^{-2}$).

## Operando Raman spectroscopy

Operando Raman spectra were collected on a high-resolution Raman spectrometer equipped with external optical path and CHI1140C electrochemical workstation. Firstly, a catalyst ink was prepared by mixing 10 mg of catalyst powder and 60 μL of Nafion in 1.0 mL of ultrapure water, which was then sprayed on a carbon paper (2.5 cm × 2.5 cm). The catalyst-coated carbon paper, Pt foil and Ag/AgCl electrode were employed as working, counter and reference electrodes, respectively. A flow-type cell with transparent module was designed to assemble above electrodes and construct three-phase interfaces for exposing to laser light. All of the detected spectra were collected with 532 nm laser wavenumber in the same working environment.

## Prototype device assembly

The device was assembled into a standalone box (size: 15 cm × 15 cm × 19 cm) composed of battery power, gas pump, two peristaltic pumps, electrolysis cell and two flasks for holding electrolytes. The electrocatalytic performances of this prototype device have been tested in natural air. To explore the potential utilization for degradation experiment, rhodamine B (Concentration: 0.03 mg mL$^{-1}$) was added into cathodic electrolyte during the operation of prototype device.

## Data availability

The datasets generated and analyzed during the present study are included in the paper and supplementary information. Source data are provided upon request.

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

## Acknowledgements
The authors would like to acknowledge the financial and equipment support from National Natural Science Foundation (Grant No. 92163124), Jiangsu Natural Science Foundation (Grant No. BK 20230097), Fundamental Research Funds for the Central Universities (Grant No. 30921013103) and the BL14W1-XAFS beamline at the Shanghai Synchrotron Radiation Facility.

## Author contributions
S.C. supervised the project and designed the experiments. Q.H., M.L., and G.H.X. performed experiments and DFT calculations. B.K.X. performed techno-economic analyses. M.A. and all authors discussed the results and assisted with the paper preparation.

## Funding

## Competing interests
Sheng Chen has filed Chinese provisional patent applications (No. 2023106808923) based on this work. Other authors declare no competing interests.
