## [Peer Review File · Nature Communications]

Beyond Le Chatelier principle: single-zinc vacancy unlocks high-rate H₂O₂ electrosynthesis from mixed dioxygenREVIEWER COMMENTS

Reviewer #1 (Remarks to the Author):

In this paper, the authors reported modified zinc oxide catalysts showing no dependence of oxygen electroreduction rates on O₂ concentrations. They clarified an updated Eley-Rideal mechanism that was verified by operando Raman spectra, temperature programmed desorption (TPD) and density function theory (DFT) calculations. Further, they conducted techno-economic analyses to simulate the scalable conditions and economic benefits of H₂O₂ electrosynthesis. After reviewing the manuscript, I recommend its publication after addressing the following concerns.

- 1) The structure figure for ER-ZnO needs to be adjusted in supplementary information. The zinc defects are not presented clearly. The local structures around the defect are not well displayed.
- 2) In Figure 4e, the authors wrote that "the ER-ZnO prefers Eley-Rideal mechanism with an energy barrier of only -0.67 eV as compared to Langmuir-Hinshelwood mechanism (0.75 eV)." Can the energy barrier be a negative value? The authors should fix this error and recheck the whole manuscript carefully.
- 3) The authors calculated the adsorption energies of *O₂, *N₂ and *O₂-*H on catalyst surfaces. Can the authors further confirm the updated Eley-Rideal mechanism for ER-ZnO based on adsorption energies?
- 4) In general, the OOH* is a key intermediate in oxygen reduction reaction. The authors should compare the *OOH adsorption energies for different samples, and illustrate its significance for reaction activities.
- 5) From the demonstration of Gibbs free energy change, the authors identify that ER-ZnO prefers the Eley-Rideal mechanism while the reference ZnO Langmuir-Hinshelwood mechanism. But the in-depth insight into the understanding of these mechanisms is not clear. They should give further discussions of these mechanisms for oxygen electroreduction.

Reviewer #2 (Remarks to the Author):

The authors mainly discussed a model system consisting of ER-ZnO with single zinc vacancies, which exhibited a violation of Le Chatelier's principle for O₂ electroreduction to H₂O₂ in mixed dioxygen. They observed that the ER-ZnO exhibited high H₂O₂ yields (25.89~24.99 mol g⁻¹ h⁻¹) and high Faradaic efficiencies (92.5%~89.3%) in the O₂ levels of 100%~21% for both the RRDE and flow cell. The improvement in selectivity was attributed to the transition from the conventional Langmuir-Hinshelwood mechanism to an updated Eley-Rideal mechanism. They further verified the Eley-Rideal mechanism through both experimental and theoretical means. These results are interesting. However, some details still need to be clarified.

1. Some crystallographic data for ZnO, as indicated in HRTEM images, were not included for ER-ZnO.
2. The XRD patterns of ER-ZnO closely resembled those of the reference ZnO. Can the absence of zinc atoms be detected through conventional XRD analysis? The authors should carefully review it by restating the characterizations or citing relevant literature.
3. In the XAFS section, the author claims that ER-ZnO has a lower Zn-O coordination number compared to the reference ZnO. Does this indicate the presence of an oxygen vacancy? If so, what impact does it have on the catalytic mechanism?
4. The authors only discussed the Zn K-edge XAFS in relation to atomic and electronic structures. Were there any changes in the surrounding oxygen atoms due to zinc vacancies? The XAFS data for oxygen atoms requires further discussion.

5. The authors are encouraged to provide a comprehensive discussion of XAFS data, for example by elucidating the connection between electron structure and electrochemical activities.

6. According to the authors' test results, the dosage of glycerol had a significant impact on the experimental outcomes. It means that the inclusion of glycerol in the synthesis process is crucial to the creation of zinc vacancies, which could be pivotal in the transition from the Langmuir-Hinshelwood to the Eley-Rideal mechanism. This requires detailed explanations and references to relevant literature.

7. Figs. 1 and 3 indicate that the decreased electron transfer in ER-ZnO is due to the successful prevention of the 4-electron process that generates H₂O, thereby effectively enhancing the production of H₂O₂. The author should explain why ER-ZnO can improve the selectivity of the H₂O₂ pathway and inhibit the four-electron pathway. In addition, the DFT calculation should compare and analyze the selectivity and selection mechanism of ER-ZnO for both the two- and four-electron pathways.

8. The authors have conducted techno-economic analyses for the electrosynthesis of H₂O₂ through the oxygen reduction reaction. However, the stability test in the experimental section was only 30 hours, which is insufficient to demonstrate the potential for scalable conditions. Similar to the literature, the authors should demonstrate the stability of the electrode for at least 100 hours.

9. The advantages and disadvantages of the proposed single-zinc vacancy catalyst should be compared to those of traditional single-atom catalysts.

Reviewer #3 (Remarks to the Author):

This work demonstrated that Zn defects in ZnO nanoplates promote selective reaction toward the two-electron oxygen reduction reaction (2e⁻ ORR) for H₂O₂ electrosynthesis under O₂-deficient gas feed conditions (i.e., air). The higher H₂O₂ selectivity with air feed on defected ZnO (referred to as ER-ZnO) compared to defect-less ZnO was confirmed using both a typical rotating ring disk electrode (RRDE) system and a flow cell employing a gas-diffusion electrode. Technoeconomic analysis for the H₂O₂ electrosynthesis using air was performed.

This reviewer acknowledges the practical significance of the efficient H₂O₂ electrosynthesis using air and the inclusion of a cost analysis. However, I found less correlation between the experimental support and the electrocatalytic performances. In particular, expressions such as "beyond the Le Chatelier principle" are considered overly exaggerated and lack verification within the study (detailed comments provided below). Therefore, this reviewer does not recommend the publication of this paper in Nature Communications.

1. The Le Chatelier principle states how chemical equilibrium changes in response to factors such as concentration and temperature. This means that this principle is related to the equilibrium potential of the reaction, which can be expressed by Nernst equation. Decrease in the O₂ partial pressure results in the decrease in the equilibrium potential for the 2e⁻ ORR. The calculation by this reviewer suggested that the onset potential would be lowered by 0.04 V when the O₂ partial pressure decreased to 20%. As expected from the calculation, the onset potential in the RRDE test significantly decreased for both ZnO and ER-ZnO. In the flow cell tests, the difference of 0.04 V is too small to be seen in the linear sweep voltammetry curves in Figure 1a and 3a. Therefore, the expression "beyond the Le Chatelier principle" is an overstatement.

2. The O₂ partial pressure has significant effects on the kinetics, not only on thermodynamics as stated by the Nernst equation. For example, it is well known that the ORR follows the first order kinetics with respect to the O₂ concentration. The O₂ partial pressure can impact on the surface coverage of the reaction intermediates, which in turn affect the adsorption properties of the catalysts. The increase in the 4e⁻ ORR selectivity under low O₂ pressure for defect-less ZnO is intriguing, implying that the *OOH adsorption strengthened.

3. The competitive adsorption of O₂ or N₂ may be important in practical application of H₂O₂ electrosynthesis technology using ambient air as a feed gas. In this regard, the authors proved O₂-selective adsorption property of ER-ZnO via TPD and DFT. However, the adsorption of N₂ under the ORR potentials is rarely discussed in most previous studies. This is because the N₂ activation and adsorption requires a large negative overpotential and thus occurs far below 0 V (vs. RHE), which is not the potential range in which the electrochemical tests are performed in this paper. If the O₂-selective adsorption property of ER-ZnO is a possible reason for its higher H₂O₂ selectivity in air, this statement should be more rigorously supported.

4. In addition to the previous comment, the Eley-Rideal mechanism has been also rarely discussed in this field. Typically, it is well accepted that the formation of *OOH intermediate during the ORR is protonation of *O₂⁻. Of course, it depends on the electrolyte conditions and catalyst types whether the protonation is followed by electron transfer from catalyst to O₂ or is concerted with the electron transfer (L-H mechanism). Therefore, the large gap of the calculated free energy between E-R and L-H mechanism is just a demonstration of the generally accepted ORR mechanism.

5. The in situ Raman signals for *O₂ and *OOH are hardly differentiated from the background noises, which makes the related discussion questionable.

List of changes

- (1) We have clearly marked the defects of ER-ZnO (**Response to Reviewer #1 Comment 1**; Supplementary Figure 23)
- (2) The term “energy barrier” has been changed to “Gibbs free energy change” to provide a more accurate description of calculation results. (**Response to Reviewer #1 Comment 2**; Page 10 line 25-Page 11 line 7).
- (3) We have discussed the significance of adsorption energies in mechanism study (**Response to Reviewer #1 Comment 3**; Page 9 line 21- page 10 line 9)
- (4) We have supplemented and discussed *OOH adsorption energy data for ER-ZnO and ZnO (**Response to Reviewer #1 Comment 4**; Figure 4d; Page 9 line 21-page 10 line 2)
- (5) We have further discussed the underlying catalytic mechanisms for ER-ZnO and ZnO based on experiments and theoretical simulations. (**Response to Reviewer #1 Comment 5**; Supplementary Figure 36; Page 8 line 15-Page 9 line 3)

Response to Reviewer #1

Original Comment: *In this paper, the authors reported modified zinc oxide catalysts showing no dependence of oxygen electroreduction rates on O_2 concentrations. They clarified an updated Eley-Rideal mechanism that was verified by operando Raman spectra, temperature programmed desorption (TPD) and density function theory (DFT) calculations. Further, they conducted techno-economic analyses to simulate the scalable conditions and economic benefits of H_2O_2 electrosynthesis. After reviewing the manuscript, I recommend its publication after addressing the following concerns.*

Original Comment 1. *The structure figure for ER-ZnO needs to be adjusted in supplementary information. The zinc defects are not presented clearly. The local structures around the defect are not well displayed.*

Response: Thanks for your kind reminding. In the revised paper, the defects of ER-ZnO have been marked, and the surrounding structures of the defects have been magnified for clear visualization as follows (Please also see attached files of ER-ZnO.cif and ZnO.cif for more structural details):

Supplementary Figure 23 The local structures of ER-ZnO, where the zinc defects are clearly marked.

Original Comment 2. *In Figure 4e, the authors wrote that “the ER-ZnO prefers Eley-Rideal mechanism with an energy barrier of only -0.67 eV as comparison to Langmuir-Hinshelwood mechanism (0.75 eV).” Can the energy barrier be a negative value? The authors should fix this error and recheck the whole manuscript carefully.*

Response: We are sorry for making the mistake. In this work, we have calculated the free energy of each intermediate according to the equation: $\Delta G = \Delta H - T\Delta S$. Then

the free energies of different intermediates have been plotted following the reaction pathway in Figure 4e. Therefore, the “-0.67 eV” represents the free energy change between different intermediates, indicating the reaction step in Eley-Rideal mechanism a spontaneous process. In the revised manuscript, the term “energy barrier” has been changed to “Gibbs free energy change” to provide a more accurate description of the reaction process. We have further rechecked the whole manuscript carefully, and revised relevant sections as follows (Page 10 line 25-Page 11 line 7 in the main text):

“As revealed in Figures 4e-g, the ER-ZnO prefers E-R mechanism with a free energy change of only -0.67 eV as comparison to L-H mechanism mechanism (0.75 eV). This result is supported by the transition state computations of E-R mechanism for ER-ZnO by a climbing image nudged elastic band method with implicit solvation model (supplementary Figure 37), showing excellent reaction kinetics with the transition energy gap of only 0.057 eV (supplementary Figure 37). In great contrast, the reference ZnO counterpart tends to proceed *via* L-H mechanism mechanism rather than E-R mechanism under the same computation condition (free energy change of 0.37 vs. 1.28 eV, supplementary Figure 38-41).” (Page 10 line 25-Page 11 line 7)

Original Comment 3. *The authors calculated the adsorption energies of *O₂, *N₂ and *O₂-*H on catalyst surfaces. Can the authors further confirm the updated Eley-Rideal mechanism for ER-ZnO based on adsorption energies?*

Response: Thanks for your useful suggestions. Yes, these adsorption energies can provide additional evidence for updated Eley-Rideal (E-R) mechanism.

Specifically, the adsorption energies of *O₂ and *N₂ has been calculated as -1.00 and -0.34 eV, showing the high selectivity of O₂ by ER-ZnO, which is necessary for ORR proceeding in mixed O₂/N₂ atmosphere. Next, the intermediate *O₂-*H plays a crucial role in the Langmuir-Hinshelwood mechanism. Upon analyzing the adsorption energies, we have found the adsorption energy of *O₂-*H is positive (0.59 eV) for ER-ZnO, indicating the stabilization of *O₂-*H intermediate in Langmuir-Hinshelwood mechanism is challenging. This is different from the negative

adsorption energy on normal ZnO surface with Langmuir-Hinshelwood mechanism ($*O_2$ - $*H$ adsorption energy: -3.63 eV). Therefore, ER-ZnO prefers Eley-Rideal mechanism while ZnO prefers Langmuir-Hinshelwood mechanism.

Based on above discussions, the following sections have been revised (Page 9 line 21- page 10 line 9):

“The adsorption energy levels on ER-ZnO are -1.00 eV for $*O_2$, -1.51 eV for $*OOH$, -0.34 eV for $*N_2$ and 0.59 eV for $*O_2$ - $*H$ (positive value indicative of unstable adsorption), and in accordance the reference ZnO are -3.77 eV for $*O_2$, -2.62 eV for $*OOH$, -1.25 eV for $*N_2$ and -3.63 eV for $*O_2$ - $*H$, respectively (Figure 4d). Different from the reference ZnO favorably adsorbing all the species of O_2 , $*OOH$, N_2 and protons, the ER-ZnO only binds stably with O_2 and $*OOH$. Appropriate $*OOH$ adsorption indicates it stable on the surface, leading to high reaction activity for two-electron ORR. Further, ER-ZnO shows high selectivity to O_2 by displaying larger binding energy relative to N_2 (-1.00 vs. 0.34 eV), which is necessary for ORR proceeding in mixed O_2/N_2 atmosphere. Next, the intermediate $*O_2$ - $*H$ plays a crucial role in the L-H mechanism. Upon analyzing the adsorption energies, the adsorption energy of $*O_2$ - $*H$ is positive (0.59 eV), indicating the stabilization of $*O_2$ - $*H$ intermediate in L-H mechanism is challenging on the surface of ER-ZnO. This is different from the negative adsorption energy on reference ZnO surface with L-H mechanism ($*O_2$ - $*H$ adsorption energy: -3.63 eV). Therefore, ER-ZnO prefers E-R mechanism while ZnO prefers L-H mechanism.” (Page 9 line 21- page 10 line 9)

Original Comment 4. *In general, the OOH^* is a key intermediate in oxygen reduction reaction. The authors should compare the $*OOH$ adsorption energies for different samples, and illustrate its significance for reaction activities.*

Response: We would like to thank the reviewer for his/her useful suggestion. The $*OOH$ adsorption energy data for ER-ZnO and ZnO have been illustrated in Figure 4d and discussed as follows:

As pointed out by the reviewer, $*OOH$ is a crucial intermediate in the two-electron ORR, where its adsorption strength directly impacts the reaction activity.

According to Sabatier's principle, strong or weak adsorption of the key *OOH intermediate can adversely affect the reaction activities. If the adsorption of *OOH is too strong, the O-O bond will break, leading to the reaction favorable of four-electron pathway. On the other hand, if the *OOH adsorption strength is too weak, the intermediate cannot be stabilized, resulting in low reaction activity. Consequently, appropriate adsorption energy indicates the *OOH stable on the catalyst surface, leading to high reaction activity toward two-electron ORR.

Based on our calculation data in Figure 4d, the adsorption energy of *OOH on ER-ZnO is appropriate (-1.51 eV), indicating the intermediate stable on the surface of ER-ZnO. On the other hand, the *OOH adsorption energy on ER-ZnO is not as high as ZnO (-2.62 eV). Therefore, we conclude ER-ZnO prefers two-electron ORR rather than four-electron ORR, which agrees well with the experimental data.

Based on above discussions, the following changes have been made in revised manuscript (Figure 4d; Page 9 line 21-page 10 line 2)

Figure 4d. The adsorption energies of different reaction intermediates on ER-ZnO and ZnO.

“The adsorption energy levels on ER-ZnO are -1.00 eV for *O₂, -1.51 eV for *OOH, -0.34 eV for *N₂ and 0.59 eV for *O₂-*H (positive value indicative of unstable adsorption), and in accordance the reference ZnO are -3.77 eV for *O₂, -2.62 eV for *OOH, -1.25 eV for *N₂ and -3.63 eV for *O₂-*H, respectively (Figure 4d). Different from the reference ZnO favorably adsorbing all the species of O₂, *OOH, N₂ and protons, the ER-ZnO only binds stably with O₂ and *OOH. Appropriate *OOH

adsorption indicates it stable on the surface, leading to high reaction activity for two-electron ORR.” (Page 9 line 21-page 10 line 2)

Original Comment 5. *From the demonstration of Gibbs free energy change, the authors identify that ER-ZnO prefers the Eley-Rideal mechanism while the reference ZnO Langmuir-Hinshelwood mechanism. But the in-depth insight into the understanding of these mechanisms is not clear. They should give further discussions of these mechanisms for oxygen electroreduction.*

Response: We thank the reviewer for his/her the constructive comment. Accordingly, we have provided more in-depth discussions on the reaction mechanism as follows:

According to the literature,^{Nat. Chem. 11, 722-729 (2019)} classical Langmuir-Hinshelwood (L-H) mechanism can be described as follows (Supplementary Figure 36): both reactant gas molecules (A and B) need to be adsorbed on the surface of catalyst (*A and *B), followed by the coupling of adjacent *A and *B to form the final product of *AB. On the other hand, the Eley-Rideal (E-R) mechanism only requires the adsorption of a single reactant gas molecule (*e.g.*, A), followed by *A on the surface combining with the free gaseous molecule B in the surrounding environment to form the final product of *AB. In this paper, we have proposed an updated E-R mechanism according to the source of reactants: gaseous A is firstly adsorbed on the catalyst surface to form *A, followed by the free species B in the liquid phase directly combining with *A at triple-phase interface, finally forming the product of *AB.

We have investigated the changes in surface intermediates through *operando* Raman experiments, which shows the characteristic peaks for *O₂ and *OOH in response to applied potentials. The peak intensities of *O₂ and *OOH have been analyzed in Figure 4c. For L-H mechanism, the proton occupies part of active sites, resulting in a relatively low concentration of *O₂ on surface and consequently a low peak intensity (for reference ZnO). In contrast, the E-R mechanism allows for more active sites for the adsorption of O₂, resulting in a higher *O₂ peak intensity (for ER-ZnO).

Further examination of ER-ZnO reveals the tendency of $*O_2$ peak intensity decrease while $*OOH$ increase with elevated applied potentials. This is originated from E-R mechanism that causes the direct coupling between $*O_2$ and dissociative H in the electrolyte, resulting in a rapid reaction rate than $*OOH$ protonation to produce H_2O_2 . Due to the sufficient supply of $*O_2$ species in E-R mechanism, $*OOH$ continuously accumulates on the surface of ER-ZnO, leading to the simultaneous decrease of $*O_2$ species. In great contrast, the change of $*OOH$ peak intensity shows the similar tendency to $*O_2$ peak intensity on the ZnO surface. This is originated from L-H mechanism, where $*O_2$ binds to $*H$ before generating $*OOH$, resulting in a parallel change of these two intermediates.

The above result is consistent to temperature programmed desorption (TPD) in N_2/O_2 mixed atmosphere and the calculated adsorption energies of oxygen-containing species on the surface of the catalysts. In the TPD results, ER-ZnO exhibits a higher selective adsorption capacity for O_2 as compared to ZnO, indicating the concentration of $*O_2$ species on ER-ZnO surface overtaking that of ZnO, which is more favorable for coupling with protons to produce $*OOH$. The adsorption energy calculations show instable $*O_2$ - $*H$ adsorption on the surface of ER-ZnO, indicating it unfavorable for L-H mechanism. Therefore, ER-ZnO prefers E-R mechanism for two-electron ORR.

Finally, we calculated the free energy diagrams of ER-ZnO and ZnO during two-electron ORR. For ER-ZnO, the E-R mechanism pathway ($*O_2 \rightarrow *OOH$) has a negative free energy change and thus tends to react spontaneously; while the L-H mechanism pathway ($*O_2 \rightarrow *O_2-*H \rightarrow *OOH$) has a positive free energy change, indicating a high reaction energy barrier precluding it to occur. For ZnO, such high-energy barrier process has been split into two low-energy change processes, making the reaction easier to proceed. Further transition state search supports that the E-R mechanism for ER-ZnO ($*O_2 \rightarrow *OOH$) is not a completely spontaneous reaction and still requires an additional energy input (applied potential) across a smaller energy barrier.

Based the above comprehensive discussions, we have added the following sentences and figures into revised manuscript (Supplementary Figure 36; Page 8 line

15-Page 9 line 3):

Supplementary Figure 36. The schematic diagrams of L-H mechanism, E-R mechanism and triple-phase E-R mechanism.

“The peak intensities of $*O_2$ and $*OOH$ have been analyzed in Figure 4c. The changes in surface intermediates shows the characteristic peaks for $*O_2$ and $*OOH$ in response to applied potentials. For L-H mechanism, the proton occupies part of active sites, resulting in a relatively low concentration of $*O_2$ on surface and consequently a low peak intensity (for reference ZnO). In contrast, the E-R mechanism allows for more active sites for the adsorption of O_2 , resulting in a higher $*O_2$ peak intensity (for ER-ZnO). Further examination of ER-ZnO reveals the tendency of $*O_2$ peak intensity decrease while $*OOH$ increase with elevated applied potentials. This is originated from E-R mechanism that causes the direct coupling between $*O_2$ and dissociative H in the electrolyte, resulting in a rapid reaction rate than $*OOH$ protonation to produce H_2O_2 . Due to the sufficient supply of $*O_2$ species in E-R mechanism, $*OOH$ continuously accumulates on the surface of ER-ZnO, leading to the simultaneous decrease of $*O_2$ species. On the other hand, the change of $*OOH$ peak intensity shows the similar tendency to $*O_2$ peak intensity on the ZnO surface. This is originated from L-H mechanism, where $*O_2$ binds to $*H$ before generating $*OOH$, resulting in a parallel change of these two intermediates.” (Page 8 line 15-Page 9 line 3)

List of changes

- (1) We have further marked the HR-TEM image of ER-ZnO. (**Response to Reviewer #2 Comment 1**; Supplementary Figure 3)
- (2) We have re-tested the XRD of ER-ZnO and discussed the XRD data. (**Response to Reviewer #2 Comment 2**; Figure R1; Page 5 line 9-11).
- (3) We have thoroughly analyzed the XAFS data that shows the absence of oxygen vacancies. (**Response to Reviewer #2 Comment 3**; Page 5 line 11-13; Page 6 line 6-8; Reference 20)
- (4) We have carefully analysed O K-edge XANES of ER-ZnO and ZnO. (**Response to Reviewer #2 Comment 4 and Comment 5**; Supplementary Figure 8; Page 6 line 1- Page 7 line 5; Reference 22, 24-27)
- (5) We have discussed the role of glycerol in mechanism study. (**Response to Reviewer #2 Comment 6**; Page 5 line 2-5; Reference 19)
- (6) We have further calculated the free energy diagrams of four-electron ORR on ER-ZnO and ZnO. (**Response to Reviewer #2 Comment 7**; Supplementary Figure 40-41; Supplementary Note 6; Supplementary information Reference 30)
- (7) We have conducted 100-hour stability test at 200 mA cm⁻². (**Response to Reviewer #2 Comment 8**; Figure 3g; Page 8 line 3-5; Supplementary Table 4)
- (8) We have discussed the advantages and disadvantages of single-zinc vacancy catalyst as comparison to traditional single-atom catalysts. (**Response to Reviewer #2 Comment 9**; Supplementary Note 1; Supplementary information Reference 16-20)

Response to Reviewer #2

Original Comment: *The authors mainly discussed a model system consisting of ER-ZnO with single zinc vacancies, which exhibited a violation of Le Chatelier 's principle for O₂ electroreduction to H₂O₂ in mixed dioxygen. They observed that the ER-ZnO exhibited high H₂O₂ yields (25.89~24.99 mol gcat⁻¹ h⁻¹) and high Faradaic efficiencies (92.5%~89.3%) in the O₂ levels of 100%~21% for both the RRDE and flow cell. The improvement in selectivity was attributed to the transition from the conventional Langmuir-Hinshelwood mechanism to an updated Eley-Rideal mechanism. They further verified the Eley-Rideal mechanism through both experimental and theoretical means. These results are interesting. However, some details still need to be clarified.*

Original Comment 1. *Some crystallographic data for ZnO, as indicated in HRTEM images, were not included for ER-ZnO.*

Response: Thanks for your kind reminding. The HR-TEM images of ER-ZnO have been further analyzed with crystallographic data marked as follows:

Supplementary Figure 3. Structural characterizations of the ER-ZnO. a, b, HR-TEM image. **c,** FFT image of HR-TEM. **d,** SAED image of HR-TEM. **e, f,** The intensity profile of (100) and (101) lattice plane.

Original Comment 2. *The XRD patterns of ER-ZnO closely resembled those of the reference ZnO. Can the absence of zinc atoms be detected through conventional XRD analysis? The authors should carefully review it by restating the characterizations or citing relevant literature.*

Response: We thank the reviewers for his/her useful suggestion. Accordingly, we have re-tested the XRD of ER-ZnO (Figure R1), and discussed the crystal data as follows:

According to the literature^{Small Methods 6, 2100932 (2022); J. Mater. Chem. A 9, 1006-1013 (2021)}, the presence of defects in a material sometimes result in the alternations of XRD patterns in peak intensities and positions. However, the change is related to the defect percentages.^{Inorg. Chem. Front. 6, 2167-2177 (2019); J. Phys. Chem. C 116, 8707-8713 (2012)}. Upon the defect percentage below 10%, the XRD pattern generally shows unobvious change.

In this paper, the Zn defect percentage in ER-ZnO has been determined as 8.72% through XPS (Supplementary Figure 6). The defect sites are uniformly dispersed in the bulk phase of the material (as revealed by HR-TEM in Figure 2a), resulting in insignificant XRD pattern difference between ER-ZnO and ZnO. Particularly, only the peak intensities at some crystal surfaces in ER-ZnO (*i.e.*, (100), (002) and (101) crystal surfaces) are slightly weaker than that of ZnO.

Based on above discussions, the following sentences have been revised in the manuscript (Page 5 line 9-11):

“The unobvious difference in XRD patterns between ER-ZnO and ZnO (*i.e.*, only slight intensity alternations in (100), (002) and (101) crystal surfaces) provides additional evidence of the single-atom zinc vacancy nature.” (Page 5 line 9-11)

Figure R1. The re-tested XRD pattern of ER-ZnO.

Original Comment 3. *In the XAFS section, the author claims that ER-ZnO has a lower Zn-O coordination number compared to the reference ZnO. Does this indicate the presence of an oxygen vacancy? If so, what impact does it have on the catalytic mechanism?*

Response: Thanks for your kind reminding. Following the reviewer's comment, the XAFS data (Figure 2e-j, Supplementary Figure 7-8) has been thoroughly re-analyzed, and the relevant literature has been referred and discussed as follows:

i) XAFS data reveals the decrease in Zn-O coordination number, which is possibly due to the unsaturated structure with some atoms missing inside the material. *Angew. Chem. Int. Edit.* 59, 9171-9176 (2020) However, it is unclear whether the presence of Zn defects, O defects or both. To further quantify the defects, a series of additional characterizations have been discussed.

ii) Firstly, XPS have revealed a significantly declined percentages of Zn element as compared to O counterpart (Supplementary Figure 6), suggesting the presence of Zn defects in ER-ZnO. The fitted EXAFS data reveals the elongated Zn-Zn distance in ER-ZnO due to the absence of local Zn atoms. The decrease in Zn-Zn coordination number in ER-ZnO also supports the presence of Zn defects. *Phys. Rev. Lett.* 107, 127206 (2011); *Adv. Mater.* 34, 2106541 (2022) From the EPR test (Figure 2d), ER-ZnO exhibited a signal at $g = 2.0040$, which is attributed to metallic Zn defects. *Phys. Status Solidi (a)* 125, 571-581 (1991)

iii) On the contrary, the EXAFS (Figure 2g) show the absence of O defects in ER-ZnO, otherwise the Zn-Zn distance around the O defects should be shorter. From the EPR test (Figure 2d), the absence of a signal at $g = 2.0020$ in the EPR test curves of ER-ZnO also indicates the absence of oxygen defects.

Based on above discussions, we can conclude the presence of zinc defects while absence of oxygen defects inside ER-ZnO. Consequently, the following sentences, reference have been added into revised manuscript (Page 5 line 11-13; Page 6 line 6-8; Reference 20):

“This conclusion is verified by other characterizations: the ER-ZnO exhibits amplified electron paramagnetic resonance (EPR) peak signal at $g = 2.0040$ relative to ZnO (4.53 vs. 0.31, Figure 2d);²⁰” (Page 5 line 11-13)

“Notably, our experimental results show the absence of O defects, otherwise Zn-Zn distance around the O defects would be shortened (as also confirmed by XPS elemental analysis in Supplementary Figure 6 and EPR in Figure 2d)” (Page 6 line 6-8)

Ref 20. Pöpl, A. & Völkel, G. ESR and Photo-ESR investigations of zinc vacancies and interstitial oxygen ions in undoped ZnO ceramics. *Phys. Status Solidi (a)* **125**, 571-581 (1991).

Original Comment 4. *The authors only discussed the Zn K-edge XAFS in relation to atomic and electronic structures. Were there any changes in the surrounding oxygen atoms due to zinc vacancies? The XAFS data for oxygen atoms requires further discussion.*

Response: We are appreciated of your useful comment. We have conducted the analyses of O K-edge XANES of ER-ZnO and ZnO (Supplementary Figure 8), and given the following discussions:

In the O K-edge XANES of ER-ZnO and ZnO, the pre-edge peaks (535.1 and 537.9 eV) are attributed to the unoccupied hybridized states of O 1s electrons transitioning to Zn 3d and O 2p orbitals above the Fermi energy level, splitting into two asymmetric peaks of different energies of t_{2g} and e_g . The broad peak represents the electron transition of O 1s to hybridized orbitals of O 2p and Zn 4sp states, while the sharp peaks represent the electronic transitions of O 1s to the more localized O $2p_z$ and O $2p_{x+y}$ states. Angew. Chem. Int. Edit. 60, 22026-22034 (2021)

Notably, the peak intensity of ER-ZnO (at around 535.1 eV ~ 537.9 eV) slightly decreases due to the reduction of available empty O 2p states. This suggests more charge transfer from Zn to O atoms, resulting in an increase in Zn valence state and a decrease in O valence state. These findings are consistent to Zn K-edge XANES data.

Further, with the elevated oxidation state of Zn, the number of outer electrons in Zn atoms decreases. This can facilitate the bonding with electron-rich O atoms, which can explain the selective adsorption of O₂ by ER-ZnO in a mixed N₂/O₂ atmosphere. At the same time, due to the decrease in outer electrons, it is difficult to provide

additional electrons for bonding with O atoms. This has resulted in lower adsorption strength of O₂ on the ER-ZnO as compared to ZnO, endowing ER-ZnO with improved two-electron ORR activity.

Based on above discussions, relevant sections have been updated in the revised manuscript (Supplementary Figure 8; Page 6 line 9-23; Reference 25):

“In the O K-edge XANES of ER-ZnO and ZnO (supplementary Figure 8), the pre-edge peaks (535.1 and 537.9 eV) are attributed to the unoccupied hybridized states of O 1s electrons transitioning to Zn 3d and O 2p orbitals above Fermi energy levels, splitting into two asymmetric peaks of different energies of t_{2g} and e_g. The broad peak represents the electron transition of O 1s to hybridized orbitals of O 2p and Zn 4sp states, while the sharp peaks represent the electronic transitions of O 1s to the more localized O 2p_z and O 2p_{x+y} states.²⁵ Notably, the peak intensity of ER-ZnO (at around 535.1 eV ~ 537.9 eV) slightly decreases due to the reduction of available empty O 2p states. This suggests more charge transfer from Zn to O atoms, resulting in an increase in Zn valence state and a decrease in O valence state. These findings are consistent to Zn K-edge XANES data. Further, with the elevated oxidation state of Zn, the number of outer electrons in Zn atoms decreases, which can facilitate the bonding with electron-rich O atoms and contribute to the selective adsorption of O₂ by ER-ZnO in the mixed N₂/O₂ atmosphere. At the same time, due to the decrease in outer electrons, it is difficult to provide additional electrons for bonding with O atoms, which has resulted in smaller adsorption strength of O₂ on the ER-ZnO as compared to ZnO, endowing ER-ZnO with improved two-electron ORR activity.” (Page 6 line 9-23)

Ref 25. Li, X.-L., et al. Stabilizing transition metal vacancy induced oxygen redox by Co²⁺/Co³⁺ redox and sodium-site doping for layered cathode materials. *Angew. Chem. Int. Edit.* **60**, 22026-22034 (2021).

Supplementary Figure 8. The O k-edge XANES of ER-ZnO and ZnO.

Original Comment 5. *The authors are encouraged to provide a comprehensive discussion of XAFS data, for example by elucidating the connection between electron structure and electrochemical activities.*

Response: Following the reviewer’s comment, we have further analyzed the XAFS data, and given the following discussions on the connection between electron structure and electrochemical activities:

i) Supplementary table 2 illustrates the detailed description of the alterations in the bond length and coordination number of ER-ZnO. Specifically, the increase in the Zn-O bond length is caused by the local deletion of Zn atoms, which migrates nearby oxygen atoms to the defects and increases the Zn-O bond length. The increase in the Zn-Zn bond length is also caused by the missing of local Zn atoms, which increases the average distance between Zn atoms. The decrease in the coordination number of Zn-O and Zn-Zn is due to the Zn defects resulting from the presence of unsaturated sites in the material. Phys. Rev. Lett. 107, 127206 (2011); Adv. Mater. 34, 2106541 (2022)

ii) As mentioned in the response to comment 4, we have carefully analyzed the O k-edge XANES. The decline of the peak intensity for ER-ZnO indicates a reduction in available empty O 2p states. This has resulted in more charge transfer from the Zn atoms to the O atoms, causing the O atoms to gain more electrons.

iii) As pointed out by the reviewer, the electronic structure is closely related to

the electrochemical activity. ER-ZnO shows the increased Zn oxidation valence state and decreased number of outer electrons of the Zn atoms that facilitate bonding with electron-rich oxygen atoms. This can explain the selective adsorption of O₂ by ER-ZnO in mixed N₂/O₂ atmospheres. Further, due to the decrease of outer electrons, ER-ZnO cannot provide sufficient electrons for bonding with O atoms, resulting in appropriate adsorption strength of *O₂ as compared to ZnO. According to the Sabatier's principle,^{Nat. Commun. 15, 359 (2024)} the moderate adsorption strength for reactive species on ER-ZnO can demonstrate improved two-electron ORR activity.^{J. Am. Chem. Soc. 140, 17597-17605 (2018)}

Based on above discussions, more details on XAFS data with references have been added into the manuscript (Page 6 line 1- Page 7 line 5; Reference 22, 24-27):

“Interestingly, the increase in the Zn-O bond length (ER-ZnO: 1.968 Å vs. ZnO: 1.956 Å) is caused by the local deletion of Zn atoms, which migrates nearby O atoms to the defects and increases the Zn-O bond length. This is consistent to the increase in Zn-Zn bond length due to the missing of local Zn atoms (ER-ZnO: 3.259 Å vs. ZnO: 3.234 Å) and decrease in the coordination number of Zn-O (ER-ZnO: 3.906 vs. ZnO: 4.006) and Zn-Zn (ER-ZnO: 11.904 vs. ZnO: 12.083) due to the unsaturated Zn sites inside the material^{22, 24} Notably, our experimental results show the absence of O defects, otherwise Zn-Zn distance around the O defects would be shortened (as also confirmed by XPS elemental analysis in Supplementary Figure 6 and EPR in Figure 2d).

In the O K-edge XANES of ER-ZnO and ZnO (supplementary Figure 8), the pre-edge peaks (535.1 and 537.9 eV) are attributed to the unoccupied hybridized states of O 1s electrons transitioning to Zn 3d and O 2p orbitals above Fermi energy levels, splitting into two asymmetric peaks of different energies of t_{2g} and e_g. The broad peak represents the electron transition of O 1s to hybridized orbitals of O 2p and Zn 4sp states, while the sharp peaks represent the electronic transitions of O 1s to the more localized O 2p_z and O 2p_{x+y} states.²⁵ Notably, the peak intensity of ER-ZnO (at around 535.1 eV ~ 537.9 eV) slightly decreases due to the reduction of available empty O 2p states. This suggests more charge transfer from Zn to O atoms, resulting

in an increase in Zn valence state and a decrease in O valence state. These findings are consistent to Zn K-edge XANES data. Further, with the elevated oxidation state of Zn, the number of outer electrons in Zn atoms decreases, which can facilitate the bonding with electron-rich O atoms and contribute to the selective adsorption of O₂ by ER-ZnO in the mixed N₂/O₂ atmosphere. At the same time, due to the decrease in outer electrons, it is difficult to provide additional electrons for bonding with O atoms, which has resulted in smaller adsorption strength of O₂ on the ER-ZnO as compared to ZnO, endowing ER-ZnO with improved two-electron ORR activity.

We find the electronic structure closely related to electrochemical activities. ER-ZnO shows the increased Zn oxidation valence state and decreased number of outer electrons of the Zn atoms that facilitate bonding with electron-rich oxygen atoms. This can promote the selective adsorption of O₂ by ER-ZnO in mixed N₂/O₂ atmosphere. Further, due to the decrease of outer electrons, ER-ZnO cannot provide sufficient electrons for bonding with O atoms, resulting in appropriate adsorption strength of *O₂ as compared to ZnO. According to the Sabatier's principle,²⁶ the moderate adsorption strength for reactive species on ER-ZnO can demonstrate improved two-electron ORR activity.²⁷ (Page 6 line 1- Page 7 line 5)

Ref 22. Zhou, Z., et al. Cation vacancy enriched nickel phosphide for efficient electrosynthesis of hydrogen peroxides. *Adv. Mater.* **34**, 2106541.

Ref 24. Ciatto, G., et al. Evidence of cobalt-vacancy complexes in Zn_{1-x}Co_xO dilute magnetic semiconductors. *Phys. Rev. Lett.* **107**, 127206 (2011).

Ref 25. Li, X.-L., et al. Stabilizing transition metal vacancy induced oxygen redox by Co²⁺/Co³⁺ redox and sodium-site doping for layered cathode materials. *Angew. Chem. Int. Edit.* **60**, 22026-22034 (2021).

Ref 26. Chen, Z. W., et al. Unusual Sabatier principle on high entropy alloy catalysts for hydrogen evolution reactions. *Nat. Commun.* **15**, 359 (2024).

Ref.27. Kuo, D.-Y., et al. Measurements of oxygen electroadsorption energies and oxygen evolution reaction on RuO₂(110): A discussion of the Sabatier principle and its role in electrocatalysis. *J. Am. Chem. Soc.* **140**, 17597-17605 (2018).

Original Comment 6. *According to the authors' test results, the dosage of glycerol had a significant impact on the experimental outcomes. It means that the inclusion of glycerol in the synthesis process is crucial to the creation of zinc vacancies, which could be pivotal in the transition from the Langmuir-Hinshelwood to the Eley-Rideal mechanism. This requires detailed explanations and references to relevant literature.*

Response: Thanks for your kind reminding. Yes, glycerol is a common chemical reagent used in tuning the properties of nanomaterials. ^{Catalysts 10, 1406 (2020)} In our work, glycerol has been used as a modifier to synthesize zinc glycerate. During the calcinations in air, a significant amount of O elements escapes from the zinc glycerate. Due to the strong interaction between Zn and O, a small amount of Zn elements also escapes from the material, resulting in structural Zn defects. The change of glycerol percentages can result in a certain proportion of Zn defects in ER-ZnO.

Our characterizations have demonstrated the presence of structural Zn defects through XPS (Supplementary Figure 6), EPR (Figure 3d), XAFS (Figure 3e-j, Supplementary Figure 7-8) and others. Further, we have conducted a series of experiments and theoretical simulations, including operando Raman (Figure 4a-c), TPD (Supplementary Figure 18) and DFT calculations (Figure 4d-g), which confirmed the transformation of the reaction mechanism on the ER-ZnO surface.

It should be mentioned that the alteration of reaction mechanisms is a result of the modification in structural Zn defects. The key novelty of this work is to disentangle the close connection between the reaction feedstock and the products by regulating the catalyst structures, thereby surpassing the constraints of Le Chatelier principle.

Based on above discussions, more sentences and reference have been added into the manuscript (Page 5 line 2-5; Reference 19):

“The glycerol was used as a modifier to synthesize zinc glycerate precursor.¹⁹ During the calcination in air, a significant amount of O elements evaporated from zinc glycerate to form ZnO. Due to the strong interaction between Zn and O, some Zn elements also escaped from the material, resulting in structural Zn defects. The change of glycerol percentages can manipulate Zn defects in ER-ZnO.” (Page 5 line

2-5)

Ref 19. Cristino, A. F., et al. Glycerol role in nano oxides synthesis and catalysis. *Catalysts* **10**, 1406 (2020).

Original Comment 7. *Figs. 1 and 3 indicate that the decreased electron transfer in ER-ZnO is due to the successful prevention of the 4-electron process that generates H₂O, thereby effectively enhancing the production of H₂O₂. The author should explain why ER-ZnO can improve the selectivity of the H₂O₂ pathway and inhibit the four-electron pathway. In addition, the DFT calculation should compare and analyze the selectivity and selection mechanism of ER-ZnO for both the two- and four-electron pathways.*

Response: Thanks for your useful suggestions. Generally, the two-electron pathway proceeds as: $*O_2 \rightarrow *OOH \rightarrow H_2O_2$, while the four-electron pathway as: $*O_2 \rightarrow *OOH \rightarrow *O \rightarrow *OH \rightarrow H_2O$. Firstly, our structural optimization by theoretical simulations shows that $*O_2$ on ER-ZnO is a linear adsorption style due to the presence of defects, while the bridging adsorption style on ZnO surface (Supplementary Figures 24-25). According to the literature,^{Chem. Rev. 118, 2302-2312 (2018)} the linear adsorption style favors two-electron ORR pathway, while bridging adsorption style favorable for four-electron pathway.

Secondly, according to the Sabatier's principle, suitable adsorption energy is crucial for reaction selectivity. Strong or weak adsorption of the key $*OOH$ intermediate can adversely affect the ORR activities. If the adsorption of $*OOH$ is too strong, the O-O bond will break, leading to the reaction favoring the four-electron pathway. On the other hand, if the $*OOH$ adsorption strength is too weak, the intermediate cannot be stabilized, resulting in low reaction activity. Based on our calculation data in Figure 4d, the adsorption energy of $*OOH$ on ER-ZnO is appropriate (-1.51 eV), indicating the intermediate stable on the surface of ER-ZnO. On the other hand, the $*OOH$ adsorption energy on ER-ZnO is not as strong as ZnO (-2.62 eV). Therefore, we conclude ER-ZnO prefers two-electron ORR while ZnO prefers four-electron ORR, which agrees well with the experimental data

Further calculations of four-electron ORR diagrams on the ER-ZnO have been added and compared to two-electron pathways (Supplementary Figures 40-41). The data shows that four-electron ORR is difficult to occur on the ER-ZnO surface due to a large positive free energy change (0.28 eV) after the production of *OOH.

Based on above discussions, more sentences and references have been added into the revised manuscript. (Supplementary Figure 40-41; Ref 30, Supplementary Note 6)

Supplementary Figure 40. The free energy diagrams of two- and four-electron ORR pathway for ER-ZnO.

Supplementary Figure 41. The free energy diagrams of two- and four-electron ORR pathway for ZnO.

“Supplementary Note 6:

Firstly, the structural optimization by theoretical simulations (Supplementary Figures 24-25) shows that *O₂ on ER-ZnO is a linear adsorption style due to the presence of defects, while the bridging adsorption style on ZnO surface. The previous literature suggests that *O₂ linear adsorption favors the two-electron ORR route,

whereas bridge adsorption favors the four-electron route.³⁰

Secondly, according to Sabatier's principle, reaction selectivity is sensitive to the appropriate adsorption energy. The adsorption of the key *OOH intermediate can have a negative impact on ORR activities if it is either too strong or too weak. If the adsorption of *OOH is too strong, the O-O bond will break, leading to a reaction that favors the four-electron pathway. Conversely, if the *OOH adsorption strength is too weak, the intermediate cannot be stabilized, resulting in low reaction activity. According to the calculation data presented in Figure 4d, the adsorption energy of *OOH on ER-ZnO is appropriate (-1.51 eV), indicating intermediate stability on the surface of ER-ZnO. Nevertheless, the adsorption energy of *OOH on ER-ZnO is not as strong as that on ZnO (-2.62 eV). Therefore, ER-ZnO prefers two-electron ORR while ZnO prefers four-electron ORR, which is consistent with the experimental data (Figure 1, 3).

Additional calculations of four-electron ORR diagrams on ER-ZnO have been conducted (Supplementary Figures 40-41). The data indicates that four-electron ORR is less likely to occur on the ER-ZnO surface compared to ZnO due to a large positive free energy change after the production of *OOH (0.28 eV vs. -1.47 eV)."

Ref 30. Kulkarni, A., Siahrostami, S., Patel, A. & Nørskov, J. K. Understanding catalytic activity trends in the oxygen reduction reaction. *Chem. Rev.* **118**, 2302-2312 (2018).

Original Comment 8. *The authors have conducted techno-economic analyses for the electrosynthesis of H₂O₂ through the oxygen reduction reaction. However, the stability test in the experimental section was only 30 hours, which is insufficient to demonstrate the potential for scalable conditions. Similar to the literature, the authors should demonstrate the stability of the electrode for at least 100 hours.*

Response: Following the reviewer's comment, we have conducted chronoamperometric test for 100 hours at the current density of 200 mA cm⁻² in air (21%O₂), which shows insignificant change of current densities (Figure 3g). The strong durability of ER-ZnO is further verified by XPS survey after stability test,

showing seldom alternation of zinc vacancy percentage in ER-ZnO (Figure 3g; Page 8 line 3-5; Supplementary Table 4).

Figure 3g. The stability test for 100 hours at the current density of 200 mA cm⁻².

Supplementary Table 4. Zn and O element percentages in ER-ZnO after stability test

Material	Zn Atomic %	O Atomic %	Zn vacancy %
ER-ZnO	47.34	52.66	10.10

“Even under the industrial-level current densities, the ER-ZnO catalyst tested in 0.6 M K₂SO₄ has shown seldom activity degradation for 100 hrs at 200 mA cm⁻².” (Page 8 line 3-5)

Original Comment 9. *The advantages and disadvantages of the proposed single-zinc vacancy catalyst should be compared to those of traditional single-atom catalysts.*

Response: Following the reviewer’s suggestion, relevant literature has been added, *Electronchem. Energy Rev.* 2, 539-573 (2019); *Chem. Rev.* 120, 11703-11809 (2020); *Joule* 2, 1242-1264 (2018); *Nat. Commun.* 10, 234 (2019); *Nat. Rev. Chem.* 2, 65-81 (2018) and the advantages and disadvantages of the proposed single-zinc vacancy catalyst have been discussed as follows:

i) Advantages: firstly, the raw materials for synthesizing our single-zinc vacancy catalysts are easily obtainable (glycol and zinc acetate), and the synthesis conditions are mild, which has been achieved by the calcination at low temperature in air. In contrast, traditional single-atom catalysts typically require complex and severe synthetic conditions, for example, calcination at high temperature or etching in strong acids. Secondly, our single-zinc vacancy catalysts have high stability and numerous

active sites. The metal defects on the surface of metal oxides do not result in a significant decline of active sites or structural failure. In contrast, traditional single-atom catalysts consist of atomically dispersed metal sites with a low content but high work function. During the reaction process, these single metal atoms would tend to peel off or agglomerate, leading to the decay of catalytic activities.

ii) Disadvantages: our single-zinc vacancy catalyst has low atom utilization due to the fact of only metal sites on the outer surface contributing to catalyzing the reaction, while those zinc atoms in the bulk phase have not been directly involved in the reaction. In contrast, traditional single-atom catalysts are typically supported on low-dimensional materials with high surface area. This allows for highly exposed surface atoms for contacting with the reactants, leading to high atom utilizations.

Based on above discussions, more sentences and references have been added into the revised supplementary information (Supplementary Note 1; Supplementary information References 16-20):

“Supplementary Note 1:

According to relevant literature,¹⁶⁻²⁰ the advantages and disadvantages of the as-proposed single-zinc vacancy catalyst have been discussed:

i) Advantages: firstly, the raw materials for synthesizing our single-zinc vacancy catalysts are easily obtainable (glycol and zinc acetate), and the synthesis conditions are mild, which has been achieved by the calcination at low temperature in air. In contrast, traditional single-atom catalysts typically require complex and severe synthetic conditions, for example, calcination at high temperature or etching in strong acids. Secondly, single-zinc vacancy catalysts have high stability and numerous active sites. The metal defects on the surface of metal oxides do not result in a significant decline of active sites or structural failure. In contrast, traditional single-atom catalysts consist of atomically dispersed metal sites with a low content but high work function. During the reaction process, single metal atoms would tend to peel off or agglomerate, leading to the decline of active sites and consequently catalytic activities.

ii) Disadvantages: single-zinc vacancy catalyst has low atom utilization due to

the fact of only metal sites on the outer surface contributing to catalyzing the reactions, while those zinc atoms in the bulk phase have not been directly involved in the reaction. In contrast, traditional single-atom catalysts are typically supported on low-dimensional materials with high surface area. This allows for highly exposed surface atoms for contacting with the reactants, leading to high atom utilizations.”

Ref 16. Chen, Y., et al. Single-atom catalysts: Synthetic strategies and electrochemical applications. *Joule* **2**, 1242-1264 (2018).

Ref 17. Lang, R., et al. Non defect-stabilized thermally stable single-atom catalyst. *Nat. Commun.* **10**, 234 (2019).

Ref 18. Wang, A. Q., Li, J. & Zhang, T. Heterogeneous single-atom catalysis. *Nat. Rev. Chem.* **2**, 65-81 (2018).

Ref 19. Cheng, N., Zhang, L., Doyle-Davis, K. & Sun, X. L. Single-atom catalysts: From design to application. *Electrochem. Energy Rev.* **2**, 539-573 (2019).

Ref 20. Kaiser, S. K., et al. Single-atom catalysts across the periodic table. *Chem. Rev.* **120**, 11703-11809 (2020).

List of changes

- (1) We have gained in-depth understanding of Le Chatelier principle and explained the description of “beyond the Le Chatelier principle” (**Response to Reviewer #3 Comment 1**; Title; Abstract, Page 1 line 16; Conclusion, Page 14 line 17; Supplementary Table 3; Supplementary Note 2)
- (2) We have elaborated the significant effects of O₂ partial pressure on the kinetics and the selectivity for different ORR pathways. (**Response to Reviewer #3 Comment 2**; Supplementary Figure 40-41; Supplementary Figure 43-46; Supplementary Note 6-7; Supplementary information reference 31-34).
- (3) We have explained the significant role of N₂ activation in mixed atmosphere for ORR. (**Response to Reviewer #3 Comment 3**; Supplementary Figure 20; Supplementary Note 4; Supplementary information reference 26)
- (4) We have made an in-depth exploration for the proposed triple-phase E-R mechanism based on experiments and theoretical calculations. (**Response to Reviewer #3 Comment 4**; Supplementary Figure 36; Figure 4g; Supplementary Figure 39; Supplementary Note 5; Supplementary information reference 27-30)
- (5) We have conducted repetitive tests on the operando Raman to verify the scientific quality of our data. (**Response to Reviewer #3 Comment 5**; (Supplementary Figure 18-19; Figure 4c; Supplementary Note 3; Supplementary information reference 21-25)

Response to Reviewer #3

Original Comment: *This work demonstrated that Zn defects in ZnO nanoplates promote selective reaction toward the two-electron oxygen reduction reaction (2e ORR) for H₂O₂ electrosynthesis under O₂-deficient gas feed conditions (i.e., air). The higher H₂O₂ selectivity with air feed on defected ZnO (referred to as ER-ZnO) compared to defect-less ZnO was confirmed using both a typical rotating ring disk electrode (RRDE) system and a flow cell employing a gas-diffusion electrode. Technoeconomic analysis for the H₂O₂ electrosynthesis using air was performed. This reviewer acknowledges the practical significance of the efficient H₂O₂ electrosynthesis using air and the inclusion of a cost analysis. However, I found less correlation between the experimental support and the electrocatalytic performances. In particular, expressions such as “beyond the Le Chatelier principle” are considered overly exaggerated and lack verification within the study (detailed comments provided below). Therefore, this reviewer does not recommend the publication of this paper in Nature Communications.*

Original Comment 1. *The Le Chatelier principle states how chemical equilibrium changes in response to factors such as concentration and temperature. This means that this principle is related to the equilibrium potential of the reaction, which can be expressed by Nernst equation. Decrease in the O₂ partial pressure results in the decrease in the equilibrium potential for the 2e ORR. The calculation by this reviewer suggested that the onset potential would be lowered by 0.04 V when the O₂ partial pressure decreased to 20%. As expected from the calculation, the onset potential in the RRDE test significantly decreased for both ZnO and ER-ZnO. In the flow cell tests, the difference of 0.04 V is too small to be seen in the linear sweep voltammetry curves in Figure 1a and 3a. Therefore, the expression “beyond the Le Chatelier principle” is an overstatement.*

Response: We are appreciated of the reviewer for his/her helpful comment. Accordingly, we have carefully studied the basic knowledge of Le Chatelier principle through relevant literature,^{Fluid Phase Equilib. 121, 167-177 (1996)}, books^{Fermi, E. Thermodynamics. Courier}

Corporation (2012) and website (www.chemedx.org/article/moving-beyond-le-châtelier). We have gained in-depth understanding of Le Chatelier principle, and given the response as follows:

By definition, the Le Chatelier Principle is generally stated along the lines: “If a system in a state of equilibrium is disturbed, the position of equilibrium will shift in order to counteract the change”. This is a qualitative description showing how chemical equilibrium is affected by experimental parameters such as concentrations, where the chemical equilibrium constant K_c is an important indicator of Le Chatelier Principle.

Specifically, the reaction equation of two-electron ORR is as follows:

Therefore, the reaction equilibrium constant can be expressed as:

According to the Le Chatelier Principle, K_c tends to keep unchanged when the O_2 concentration decreases from 100% to 21%. Therefore, the concentration of H_2O_2 product is predicted to decrease, resulting in reduced yield rates. Indeed, the yield rate of reference ZnO catalyst shows a decline of 43.7% from 100% O_2 to 21% O_2 at the current density of 300 mA cm^{-2} . The activity decline is in accordance with common prediction from Le Chatelier Principle.

In this work, we have observed an abnormal phenomenon after introducing single-zinc vacancies into ZnO. The as-resultant ER-ZnO catalyst has shown seldom performance degradation under the same condition. Even at 300 mA cm^{-2} , the yield rates of ER-ZnO shows a decrease of $< 5\%$ from 100% to 21% O_2 . This phenomenon contradicts the prediction of Le Chatelier Principle. To illustrate such abnormal phenomenon, we describe it as “beyond the Le Chatelier Principle”. Actually, our proposed concept is only a qualitative description.

Further, according to the reviewer’s suggestion, we have compared the onset potentials of ZnO and ER-ZnO from 100% to 21% O_2 in flow cells and RRDE system (Supplementary Table 3). The results indicate the maximum onset potential changes

of ZnO as 0.036 and 0.035 V for flow cell and RRDE system, respectively. This is in compliance with the Le Chatelier Principle. While under the same condition, the maximum onset potential changes of ER-ZnO are only 0.025 and 0.014 V, respectively. This suggests a partial deviation from Le Chatelier Principle in ER-ZnO. In the manuscript, we have explained the origin of such deviation through experiments (Figure 1; Figure 3), operando spectroscopy (Figures 4a-c) and theoretical calculations (Figures 4d-g).

Collectively, our proposed concept of “beyond the Le Chatelier Principle” is only a qualitative description of the experimental results, which deviates from traditional predictions. From a more profound mechanism perspective, the reaction process still adheres to the Le Chatelier Principle. We would like to express our gratitude to the reviewers for his/her valuable comment, and we have added double quotation marks to “beyond the Le Chatelier Principle” in relevant sections to enhance the scientific contents of this description.

Based on above discussions, the following tables, sentences and references have been added into in the revised manuscript (Title; Abstract, Page 1 line 16; Conclusion, Page 14 line 17; Supplementary Table 3; Supplementary Note 2):

“Beyond the Le Chatelier principle”: single-zinc vacancy unlocks high-rate O₂ electrolysis to hydrogen peroxide in mixed dioxygen media. (Title)

allowing for leveraging systems “beyond” this classical rule. (Abstract, Page 1 line 16)

“beyond” the constraint of classical Le Chatelier principle (Conclusion, Page 14 line 17)

Supplementary Table 3. The onset potential of ER-ZnO and ZnO in flow cell and RRDE system, respectively.

Onset potentials (V)	21% O ₂	40% O ₂	80% O ₂	100% O ₂	Maximum change
ER-ZnO (Flow cells)	0.416	0.428	0.439	0.441	0.025
ZnO (Flow cells)	0.361	0.380	0.394	0.397	0.036

ER-ZnO (RRDE)	0.344	0.347	0.355	0.358	0.014
ZnO (RRDE)	0.334	0.360	0.364	0.369	0.035

“Supplementary Note 2:

The onset potential changes of ZnO are 0.036 and 0.035 V in flow cell and RRDE systems from 100% O₂ to 21% O₂, respectively. This is in compliance with the Le Chatelier Principle. While under the same condition, the onset potential changes of ER-ZnO are only 0.025 and 0.014 V, which suggests a partial deviation from Le Chatelier Principle.”

Original Comment 2. *The O₂ partial pressure has significant effects on the kinetics, not only on thermodynamics as stated by the Nernst equation. For example, it is well known that the ORR follows the first order kinetics with respect to the O₂ concentration. The O₂ partial pressure can impact on the surface coverage of the reaction intermediates, which in turn affect the adsorption properties of the catalysts. The increase in the 4e ORR selectivity under low O₂ pressure for defect-less ZnO is intriguing, implying that the *OOH adsorption strengthened.*

Response: We fully agree with the reviewer’s suggestions and give the response as follows:

i) The O₂ partial pressure indeed has significant effects on the kinetics of two-electron ORR. In electrochemical reactions, the B-V and Levich equations are generally used to describe reaction kinetics: Nano Mater. Sci. 3, 313-318 (2021).

Accordingly, the current density and O₂ concentration (C₀) have the following first order kinetics relationship:

ii) Yes, O₂ partial pressure can impact on the surface coverage of reaction intermediates. As shown in Figures 4a,b in the main text, operando Raman spectroscopy was utilized to examine the adsorption of *OOH intermediates on ZnO and ER-ZnO under 100% ~ 21% O₂ environments. The coverage of *OOH on the surface of ZnO change significantly, while on ER-ZnO remains consistent. Notably, the coverage of *OOH on ER-ZnO was greater than that of ZnO in 21% O₂. According to the literature, *J. Am. Chem. Soc.* 144, 14936-14944 (2022) the high coverage of intermediates favors a low energy barrier for the *O₂ → *OOH step, resulting in a fast reaction rate, and consequently high reaction selectivity.

To further investigate the effect of O₂ partial pressure on activities, we have tested the two-electron ORR for ER-ZnO at lower O₂ partial pressures (21% ~ 5%, Supplementary Figures 43-45). We have calculated the limiting diffusion current densities for different O₂/N₂ mixtures according to the following equation:

The symbol I_d represents the limiting diffusion current density (mA cm⁻²). The symbol \dot{V} represents the gas reactant flow rate (mol cm⁻³). D_{ON} represents the diffusion coefficient of oxygen in natural air (cm² s⁻¹), and c_N represents the nitrogen concentration at the surface of the gaseous diffusion layer (mol cm⁻³). The variables n , F and δ represent the number of electron transfer reactions, Faraday constant (96485 C mol⁻¹) and the gas diffusion layer thickness (cm), respectively.

Obviously, the limiting diffusion current densities of the reaction decrease continuously with O₂ partial pressures. Notably, our ER-ZnO catalyst can maintain the current density close to the theoretical limits under different partial pressures (Supplementary Figure 46).

Finally, as guided by the reviewer, we have examined the four-electron ORR selectivity for ZnO and ER-ZnO. Indeed, as pointed out by the reviewer: increase in four-electron ORR selectivity was observed under low O₂ pressure for defect-less ZnO. The origin of such phenomenon has been explained in the main text. Firstly, our structural optimization by theoretical simulations (Supplementary Figures 24-25)

shows that $*O_2$ on ER-ZnO is a linear adsorption style due to the presence of defects, while the bridging adsorption style on ZnO surface. According to the literature,^{Chem. Rev. 118, 2302-2312 (2018)} the linear adsorption style favors two-electron ORR pathway, while bridging adsorption style favorable for four-electron pathway. Secondly, according to the Sabatier's principle, suitable adsorption energy is crucial for reaction selectivity. Strong or weak adsorption of the key $*OOH$ intermediate can adversely affect the ORR activities. If the adsorption of $*OOH$ is too strong, the O-O bond will break, leading to the reaction favoring the four-electron pathway. On the other hand, if the $*OOH$ adsorption strength is too weak, the intermediate cannot be stabilized, resulting in low reaction activity. Based on our calculation data in Figure 4d, the adsorption energy of $*OOH$ on ER-ZnO is appropriate (-1.51 eV), indicating the intermediate stable on the surface of ER-ZnO. And the $*OOH$ adsorption energy on ER-ZnO is not as strong as ZnO (-2.62 eV). Therefore, we conclude ER-ZnO prefers two-electron ORR while ZnO prefers four-electron ORR, which agrees well with the experimental data (Figure 1, 3). Thirdly, our further calculations of four-electron ORR diagrams on the ER-ZnO and ZnO have been conducted (Supplementary Figures 40-41). The data shows that four-electron ORR is easier to occur on the ZnO surface as comparison to ER-ZnO due to a larger positive free energy change (-1.47 vs 0.28 eV).

Based on above discussions, more figures, sentences and reference have been added into the revised manuscript (Supplementary Figures 40-41; Supplementary Figures 43-46; Supplementary Notes 6-7; Supplementary information references 31-34):

Supplementary Figure 40. The free energy diagrams of two- and four-electron pathway for ER-ZnO.

Supplementary Figure 41. The free energy diagrams of two- and four-electron pathway for ZnO.

Supplementary Figure 43. The ORR activity of ER-ZnO catalyst in 5% O₂ a, H₂O₂ yields tested in flow cells. c, H₂O₂ Faradaic efficiencies tested in flow cells, inset with the corresponding H₂O₂ concentration.

Supplementary Figure 44. The ORR activity of ER-ZnO catalyst in 10% O₂ a, H₂O₂ yields tested in flow cells. c, H₂O₂ Faradaic efficiencies tested in flow cells, inset with the corresponding H₂O₂ concentration.

Supplementary Figure 45. The ORR activity of ER-ZnO catalyst in 15% O₂ a, H₂O₂ yields tested in flow cells. c, H₂O₂ Faradaic efficiencies tested in flow cells, inset with the corresponding H₂O₂ concentration.

Supplementary Figure 46. The limiting diffusion current densities of ER-ZnO catalyst in low O₂ content.

“Supplementary Note 6:

Firstly, the structural optimization by theoretical simulations (Supplementary Figures 24-25) shows that *O₂ on ER-ZnO is a linear adsorption style due to the

presence of defects, while the bridging adsorption style on ZnO surface. The previous literature suggests that $*O_2$ linear adsorption favors the two-electron ORR route, whereas bridge adsorption favors the four-electron route.³¹

Secondly, according to Sabatier's principle, reaction selectivity is sensitive to the appropriate adsorption energy. The adsorption of the key $*OOH$ intermediate can have a negative impact on ORR activities if it is either too strong or too weak. If the adsorption of $*OOH$ is too strong, the O-O bond will break, leading to a reaction that favors the four-electron pathway. Conversely, if the $*OOH$ adsorption strength is too weak, the intermediate cannot be stabilized, resulting in low reaction activity. According to the calculation data presented in Figure 4d, the adsorption energy of $*OOH$ on ER-ZnO is appropriate (-1.51 eV), indicating intermediate stability on the surface of ER-ZnO. Nevertheless, the adsorption energy of $*OOH$ on ER-ZnO is not as strong as that on ZnO (-2.62 eV). Therefore, ER-ZnO prefers two-electron ORR while ZnO prefers four-electron ORR, which is consistent with the experimental data (Figure 1, 3).

Additional calculations of four-electron ORR diagrams on ER-ZnO have been conducted (Supplementary Figures 40-41). The data indicates that four-electron ORR is less likely to occur on the ER-ZnO surface compared to ZnO due to a large positive free energy change after the production of $*OOH$ (0.28 eV vs. -1.47 eV)."

“Supplementary Note 7:

The O_2 partial pressure indeed has significant effects on the kinetics of two-electron ORR. In electrochemical reactions, the B-V and Levich equations are generally used to describe reaction kinetics.³²

Accordingly, the current density and O_2 concentration (C_0) have the following first order kinetics relationship:

Furthermore, the surface coverage of reaction intermediates can be affected by the partial pressure of O₂. As demonstrated in Figures 4a and 4b in the main text, operando Raman spectroscopy was used to examine the adsorption of *OOH intermediates on ZnO and ER-ZnO under 100% ~ 21% O₂ environments. The coverage of *OOH on the surface of ZnO change significantly, while on the ER-ZnO remains consistent. It is worth noting that the coverage of *OOH on the surface of ER-ZnO was significantly greater than that of ZnO in 21% O₂. The high coverage of intermediates, as reported in the literature³³ favors a low energy barrier for the *O₂ → *OOH step, resulting in a fast reaction rate and consequently high reaction selectivity.

To further investigate the effect of O₂ partial pressure on activities, the two-electron ORR performance of ER-ZnO were tested at lower O₂ partial pressures (21% ~ 5%, Supplementary Figure 43-45). The limiting diffusion current densities are calculated for different concentration mixtures across the electrode surface according to the following equation:³⁴

The limiting diffusion current density (I_d) (mA cm⁻²) is determined by the gas reactant flow (Γ) (mol cm⁻³), the diffusion coefficient of oxygen in natural air (D_{O_2}) (cm² s⁻¹), and the concentration of nitrogen on the surface of the gaseous diffusion layer (c_N) (mol cm⁻³). Additionally, the electron reaction numbers (n), the Faraday constant (F) (96485 C mol⁻¹), and the thickness of the gaseous diffusion layer (δ) (cm) are also considered.

Obviously, the current densities of the reaction decrease continuously with O₂ partial pressures. Notably, The ER-ZnO catalyst can maintain the current density close to the theoretical limits under different partial pressures (Supplementary Figure 46).”

Ref 31. Kulkarni, A., Siahrostami, S., Patel, A. & Nørskov, J. K. Understanding catalytic activity trends in the oxygen reduction reaction. *Chem. Rev.* **118**, 2302-2312

(2018).

Ref 32. Wang, J., et al. Quantitative kinetic analysis on oxygen reduction reaction: A perspective. *Nano Materials Science* **3**, 313-318 (2021).

Ref 33. Zheng, M., et al. Electrocatalytic CO₂-to-C₂⁺ with ampere-level current on heteroatom-engineered copper via tuning *CO intermediate coverage. *J. Am. Chem. Soc.* **144**, 14936-14944 (2022).

Ref 34. Zha, Q. X. Introduction to electrode process kinetics, Ch. 9 (Science Press, 2002).

Original Comment 3. *The competitive adsorption of O₂ or N₂ may be important in practical application of H₂O₂ electrosynthesis technology using ambient air as a feed gas. In this regard, the authors proved O₂-selective adsorption property of ER-ZnO via TPD and DFT. However, the adsorption of N₂ under the ORR potentials is rarely discussed in most previous studies. This is because the N₂ activation and adsorption requires a large negative overpotential and thus occurs far below 0 V (vs. RHE), which is not the potential range in which the electrochemical tests are performed in this paper. If the O₂-selective adsorption property of ER-ZnO is a possible reason for its higher H₂O₂ selectivity in air, this statement should be more rigorously supported.*

Response: We are appreciated of the reviewer for his/her useful comment. We have made the response as follows:

i) As pointed out by the reviewer, N₂ activation has been rarely studied in the literature, which is mainly because of the as-reported two-electron ORR studies focusing on high-purity O₂ environment (> 99.9%). In this work, we have investigated the two-electron ORR performances in mixed N₂/O₂ environments. Our findings would advance the large-scale electrochemical synthesis of hydrogen peroxide.

ii) Theoretically, O₂ and N₂ activation at the cathode mainly proceed through the following equations: Adv. Energy Mater. 8, 1800369 (2018)

Obviously, the theoretical potentials of 2e-ORR and NRR only exhibit limited gap (0.453 V), so in most cases it is necessary to take consider of both reactions.

iii) Experimentally, we have utilized flow-cell systems to achieve industrial-grade current densities (Figure 3b). In air environment (21% O₂), the applied potentials on the ER-ZnO catalyst at 50 ~ 300 mA cm⁻² are 0.21 ~ -0.67 V. On the other hand, at the current density of 200 mA cm⁻², the applied potentials on the ER-ZnO catalyst are -0.27 V (100% O₂), -0.24 V (80% O₂), -0.25 V (40% O₂) and -0.36 V (21% O₂), respectively. Therefore, both two-electron ORR and NRR may occur simultaneously under such potential range.

iv) Accordingly, we have investigated the effect of N₂ activation by using TPD, operando Raman spectra and DFT calculations. In the TPD experiments, the adsorption capacity of ER-ZnO (0.104) on N₂ was less than that of ZnO (0.167, Supplementary Figure 21). Similarly, the vibrational peaks of *N₂ were tested on operando Raman, showing much smaller peak intensity of ER-ZnO as comparison to ZnO (Supplementary Figure 20). Further, the adsorption of N₂ on the catalyst surface was calculated by using DFT, showing the smaller adsorption energy of ER-ZnO as comparison to ZnO (-0.34 vs -1.25 eV, Figure 4d).

Based on above discussions, more figures, sentences and reference have been added into the revised manuscript (Supplementary Figure 20; Supplementary Note 4; Supplementary information reference 26)

Supplementary Figure 20. The peak intensity of *N₂ in operando Raman test.

“Supplementary Note 4:

N₂ activation has been rarely studied in the literature, which is mainly because of current two-electron ORR studies focusing on high-purity O₂ environment (> 99.9%). In this work, the two-electron ORR performances was investigated in mixed N₂/O₂ environments. The findings would advance the large-scale electrochemical synthesis of hydrogen peroxide in the near future.

Generally, O₂ and N₂ activation at the cathode mainly follows the following equations:²⁶

Obviously, the theoretical potentials of 2e-ORR and NRR only shows limited gap (0.453 V), so in most cases it is necessary to consider both reactions..

Experimentally, flow-cell systems were utilized to achieve industrial-grade current densities (Figure 3b). In air environment (21% O₂), the applied potentials on the ER-ZnO catalyst at 50 ~ 300 mA cm⁻² current densities are 0.21 ~ -0.67 V. On the other hand, at the current density of 200 mA cm⁻², the applied potentials on the ER-ZnO catalyst are -0.27 V (100% O₂), -0.24 V (80% O₂), -0.25 V (40% O₂) and -0.36 V (21% O₂), respectively. Therefore, both 2e-ORR and NRR may occur simultaneously under such potential range.

Accordingly, the effect of N₂ activation is investigated by using TPD (Supplementary Figure 21), operando Raman spectra (Figure 4a-c) and DFT calculations (Figure 4d-g). In the TPD experiments, the adsorption capacity of ER-ZnO (0.104) on N₂ was less than that of ZnO (0.167, Supplementary Figure 21). Similarly, the vibrational peaks of *N₂ were tested on operando Raman, showing much smaller peak intensity of ER-ZnO as comparison to ZnO (Supplementary Figure 20). The adsorption of N₂ on the catalyst surface was calculated by using DFT, showing the smaller adsorption energy of ER-ZnO as comparison to ZnO (-0.34 vs -1.25 eV, Figure 4d).”

Ref 26. Cui, X., Tang, C. & Zhang, Q. A review of electrocatalytic reduction of

dinitrogen to ammonia under ambient conditions. *Adv. Energy Mater.* **8**, 1800369 (2018).

Original Comment 4. *In addition to the previous comment, the Eley-Rideal mechanism has been also rarely discussed in this field. Typically, it is well accepted that the formation of *OOH intermediate during the ORR is protonation of *O₂⁻. Of course, it depends on the electrolyte conditions and catalyst types whether the protonation is followed by electron transfer from catalyst to O₂ or is concerted with the electron transfer (L-H mechanism). Therefore, the large gap of the calculated free energy between E-R and L-H mechanism is just a demonstration of the generally accepted ORR mechanism.*

Response: We would like thank the reviewers for his/her valuable comment. We have made the response as follows:

i) Indeed, as pointed out by the reviewer, the Eley-Rideal (E-R) mechanism has not been extensively studied in two-electron ORR. This is primarily because of the as-reported 2e-ORR studies seldom exploring the impact of proton sources on activities. In this work, we have carefully examined the intermediate processes of ZnO and ER-ZnO for 2e-ORR. For the first time, we have found the proton sources exhibiting significant influence on activities, particularly under industrial-level current densities. Our findings may promote the industrialization of electrochemical hydrogen peroxide production.

ii) More specifically, the reaction process of 2e-ORR is usually described as follows: $O_2 + * \rightarrow *O_2 \rightarrow *OOH \rightarrow *H_2O_2 \rightarrow * + H_2O_2$. According to the literature,^{Angew. Chem. Int. Edit. 59, 9171-9176 (2020)} the first protonation process is considered as the decisive step ($*O_2 \rightarrow *OOH$), which is the focus of our study.

We have utilized heterogeneous catalysis theory to investigate the reaction mechanism of the first hydrogenation process ($*O_2 \rightarrow *OOH$). The heterogeneous catalysis theory typically involves three mechanisms: Langmuir-Hinshelwood (L-H) mechanism, Eley-Rideal (E-R) mechanism and Mars-van Krevelen (M-V) mechanism. Because of M-V mechanism mainly occurring on anion-deficient catalysts,^{Nature 583,}

391-395 (2020) we then only examine the L-H mechanism and E-R mechanism. According to the literature, ^{Nat. Chem. 11, 722-729 (2019)} classical L-H mechanism can be described as (Supplementary Figure 36): both reactant gas molecules (A and B) adsorbed on the surface of catalyst (*A and *B), followed by the coupling of adjacent *A and *B to form the final product of *AB. On the other hand, the E-R mechanism requires the adsorption of a single reactant gas molecule (*e.g.*, A), followed by *A on the surface combining with the free gaseous molecule B in the surrounding environment to form the final product of *AB.

We have noted the traditional E-R mechanism mainly occurring in solid-gas phase and is dependent on the catalyst structure and properties. ^{JACS Au 3, 943-952 (2023)} In this paper, we have updated E-R mechanism according to the source of reactant: the gaseous A is firstly adsorbed on the catalyst surface to form *A, followed by the free species B in the liquid phase directly combining with *A at triple-phase interface, finally forming the product of *AB.

We have further confirmed the updated E-R mechanism through a series of experiments (such as *operando* Raman and temperature programmed desorption, Figure 4a-c) and theoretical calculations (Figure 4d-g). Particularly in theoretical calculations, we have plotted the kinetic barrier of E-R and L-H mechanisms on ER-ZnO and ZnO. As shown in Figure 4g and Supplementary Figure 39, ER-ZnO prefers E-R than L-H mechanisms because of smaller energy barriers (-0.67 vs 0.74 eV), which is different from ZnO (1.29 vs 0.38 eV).

Based on above discussions, the following figures, sentences and references have been added into in the revised manuscript (Supplementary Figure 36; Figure 4g; Supplementary Figure 39; Supplementary Note 5; Supplementary information References 27-30):

Supplementary Figure 36. The schematic diagrams of L-H mechanism, E-R mechanism and triple-phase E-R mechanism.

Figure 4g. kinetic barriers for the hydrogenation of $*O_2$ to $*OOH$ on ER-ZnO via Eley-Rideal and Langmuir-Hinshelwood mechanisms.

Supplementary Figure 39 The kinetic barrier of E-R mechanism and L-H mechanism on the reference ZnO.

“Supplementary Note 5:

The E-R mechanism has not been extensively studied in two-electron ORR. This is primarily because of present studies seldom exploring the impact of proton sources on 2e-ORR activities. In this work, we have carefully examined the intermediate

processes of ZnO and ER-ZnO for 2e-ORR. For the first time, we have found the proton sources that have a significant influence on the 2e-ORR activities, particularly under industrial-level current densities. Our findings may promote the industrialization of electrochemical hydrogen peroxide production.

More specifically, based on the literature and our experimental results, the reaction intermediate process of 2e-ORR is as follows: $O_2 + * \rightarrow *O_2 \rightarrow *OOH \rightarrow *H_2O_2 \rightarrow * + H_2O_2$. According to the literature,²⁷ the first protonation process is considered as the decisive step ($*O_2 \rightarrow *OOH$), which is the focus of our study.

We have then utilized heterogeneous catalysis to investigate the reaction mechanism of the hydrogenation process ($*O_2 \rightarrow *OOH$). The heterogeneous catalysis theory typically involves three mechanisms: Langmuir-Hinshelwood (L-H) mechanism, Eley-Rideal (E-R) mechanism, and Mars-van Krevelen (M-V) mechanism. Because of M-V mechanism mainly occurring on anion-deficient catalysts,²⁸ we then examined the L-H mechanism and E-R mechanism. According to the literature,²⁹ classical L-H mechanism can be described as (Supplementary Figure 36): both reactant gas molecules (A and B) adsorbed on the surface of catalyst ($*A$ and $*B$), followed by the coupling of adjacent $*A$ and $*B$ to form the final product of $*AB$. On the other hand, the E-R mechanism requires the adsorption of a single reactant gas molecule (*e.g.*, A), followed by $*A$ on the surface combining with the free gaseous molecule B in the surrounding environment to form the final product of $*AB$.

We have noted the traditional E-R mechanism mainly occurring in the solid-gas phase and is dependent on the catalyst structure and properties.³⁰ In this paper, we have updated E-R mechanism according to the source of reactant: the gaseous A is firstly adsorbed on the catalyst surface to form $*A$, followed by the free species B in the liquid phase directly combining with $*A$ at triple-phase interface, finally forming the product of $*AB$.

We have further confirmed the updated E-R mechanism through a series of experiments (such as *operando* Raman and temperature programmed desorption, Figure 4a-c) and theoretical calculations (Figure 4d-g). Particularly in theoretical

calculations, we have plotted the kinetic barrier of E-R and L-H mechanisms on ER-ZnO and ZnO. As shown in Figure 4g and Supplementary Figure 39, ER-ZnO prefers E-R than L-H mechanisms because of smaller energy barriers (-0.67 vs 0.74 eV), which is different from ZnO (1.29 vs 0.38 eV).”

Ref 27. Tang, C., et al. Coordination tunes selectivity: Two-electron oxygen reduction on high-loading molybdenum single-atom catalysts. *Angew. Chem. Int. Edit.* **59**, 9171-9176 (2020)

Ref 28. Ye, T.-N., et al. Vacancy-enabled N₂ activation for ammonia synthesis on an Ni-loaded catalyst. *Nature* **583**, 391-395 (2020).

Ref 29. Liu, T., Wang, Y. & Li, Y. Can metal–nitrogen–carbon single-atom catalysts boost the electroreduction of carbon monoxide? *JACS Au* **3**, 943-952 (2023).

Ref 30. Weinberg, W. H. Eley-Rideal surface chemistry: direct reactivity of gas phase atomic hydrogen with adsorbed species. *Acc. Chem. Res.* **29**, 479-487 (1996).

Original Comment 5. *The in-situ Raman signals for *O₂ and *OOH are hardly differentiated from the background noises, which makes the related discussion questionable.*

Response: Thanks for your kind reminding. According to the reviewer's comment, we have performed additional four repetitive tests for the operando Raman data of Figures 4a,4b. These results are very close to the previous ones, so we update operando Raman results with error bars (Figure 4c). We consider the weak Raman signal due to harsh operando test conditions during experimental conditions. By further reviewing the relevant literature, we found the following reasons: *J. Phys. Chem. Lett.* **13**, 479-485 (2022); *J. Phys. D: Appl. Phys.* **57**, 103002 (2023).

Generally, Raman spectra is a scattering signal with intrinsic weak intensity. When used to study the structural characteristics of a bulk material, obvious signal peaks can be seen. This is due to the structural crystal lattice that contributes to overall vibration of the bulk material, leading to enhanced scattered signals.

While in this work, the operando Raman only probes the signals of adsorbed species during catalytic processes. The scattered Raman signals are mainly focused on

bond vibrations of adsorbed species (like $*O_2$ and $*OOH$ in 2e-ORR) on catalyst surfaces, which are known to be very weak as comparison to structural crystal lattices in bulk materials. Consequently, the operando Raman signals for catalytic reactions is mostly very weak in the literatures (like ORR, NRR, CRR).^{J. Am. Chem. Soc. 142, 715-719 (2020); ACS Nano 14, 11363-11372 (2020); Angew. Chem. Int. Edit. 60, 20331-20341 (2021)} Our operando Raman signals in Figures 4a, 4b are comparable to the above literature.

To confirm the accuracy of the experimental results, we have used Raman spectrometer software to directly determine the peak positions and intensities. We have compared the Raman information of ER-ZnO at different applied potentials. In general, the intensity of $*OOH$ peaks continue to increase while the intensity of $*O_2$ peaks decrease, which is comparable to the literature, and proves the results of our operando Raman test valid for use.

Based on above discussions, the following figures, sentences and references have been added into the revised manuscript (Supplementary Figure 18-19; Figure 4c; Supplementary Note 3; Supplementary information references 21-25):

Supplementary Figure 18. a-d, The operando Raman data of ER-ZnO in 21% O₂.

Supplementary Figure 19. a-d, The operando Raman data of ZnO in 21% O₂.

Figure 4c. The peak intensity analyses with error bar in operando Raman test.

“Supplementary Note 3:

Additional four repetitive tests were performed for the operando Raman under the same condition of Figures 4a, 4b. The weak Raman signal is due to the harsh operando test conditions.^{21, 22} In this work, the operando Raman only probes the signals of adsorbed species during catalytic processes. The scattered Raman signals are mainly focused on bond vibrations of adsorbed species (like *O₂ and *OOH in ORR) on catalyst surfaces, which are known to be very weak as comparison to bulk materials. To confirm the accuracy of the experimental results, Raman spectrometer software is used to directly determine the peak positions and intensities. On this basis, the Raman information of ER-ZnO are compared at different applied potentials. In general, the intensity of *OOH peaks continued to increase while the intensity of *O₂ peaks decrease, which is comparable to the literature,²³⁻²⁵ and proves the results of our

operando Raman test valid for use.”

Ref 21. Favaro, M., Kong, H. & Gottesman, R. In situ and operando Raman spectroscopy of semiconducting photoelectrodes and devices for photoelectrochemistry. *J. Phys. D: Appl. Phys.* **57**, 103002 (2023).

Ref 22. Li, H. Y., et al. Operando electrochemical X-ray diffraction and Raman spectroscopic studies revealing the alkali-metal ion intercalation mechanism in prussian blue analogues. *J. Phys. Chem. Lett.* **13**, 479-485 (2022).

Ref 23. Dong, J.-C., et al. Direct in situ Raman spectroscopic evidence of oxygen reduction reaction intermediates at high-index Pt(hkl) surfaces. *J. Am. Chem. Soc.* **142**, 715-719 (2020).

Ref 24. Shan, W., et al. In situ surface-enhanced Raman spectroscopic evidence on the origin of selectivity in CO₂ electrocatalytic reduction. *ACS Nano* **14**, 11363-11372 (2020).

Ref 25. Zhao, Y., et al. Identification of M-NH₂-NH₂ intermediate and rate determining step for nitrogen reduction with bioinspired sulfur-bonded FeW catalyst. *Angew. Chem. Int. Edit.* **60**, 20331-20341 (2021).

REVIEWERS' COMMENTS

Reviewer #1 (Remarks to the Author):

The authors have responded adequately to my questions and the paper can now be published.

Reviewer #2 (Remarks to the Author):

The authors have addressed all of my concerns.

Reviewer #3 (Remarks to the Author):

The authors have fully addressed this reviewer's comment with additional experiments, and this paper is now recommended for publication.